# Unlocking the return insurance puzzle in e-commerce: A strategic dance between e-sellers and the e-platform

Chen Zhang [ORCID]*

International business school, Hainan University, Haikou, China

* 694539771@qq.com

## Abstract

As return insurance has become a prevalent strategy, understanding the influences of the return insurance remains a critical question. This study considers a supply chain comprising an e-platform and two competing e-sellers with different product qualities under a commission contract. Eight duopoly game models were constructed to uncover optimal return insurance policies and their influences for e-sellers and the e-platform, considering the customer heterogeneity. Several key findings emerge:1) When the e-platform does not offer return insurance, retailers determine the size of the premium and choose the optimal return strategy based on a combination of the premium, the commission rates, the return compensation, and the return rate; 2) When the e-platform does not offer return insurance, retailers can lower their prices to encourage consumers to decide whether or not to purchase return insurance by themselves; 3) The e-platform that offers return insurance can change retailers' return strategy to the detriment of both the high quality e-sellers' profits and their own revenue.

## Introduction

To enhance consumer shopping experiences, mitigate freight losses during returns or replacements, and bolster online shopping demand, China's Alibaba Group and Huatai Insurance Co., Ltd. collaboratively introduced return freight insurance (hereinafter referred to as "return insurance") in 2011. This pioneering insurance product, initially unveiled on Taobao.com in November of that year, reimburses consumers for the return freight costs in accordance with predefined standards, regardless of whether the insurance is purchased by the consumer, e-seller, or e-platform. Boasting cost-effectiveness, efficient claims processing, and an array of other appealing attributes, return insurance swiftly captured the hearts of consumers and gained widespread popularity. Encouraged by its success, numerous companies, such as Zhongan Insurance and Guotai Property Insurance, began providing return insurance policies on Chinese e-platforms like T-mall and JD.com, contributing to a remarkable surge

**Data availability statement:** All relevant data are within the manuscript and supporting information files.

**Funding:** Postgraduate Innovative Research Projects of Hainan Province(Qhyb2022-20 and Qhyb2021-19) The funders had no role in study design, data collection and analysis, decision to publish, or preparation of the manuscript.

**Competing interests:** The authors have declared that no competing interests exist.

in China's annual return insurance premiums from a modest 170 million in 2015 to a staggering 15 billion in 2019.

However, there exists a notable disparity in the adoption of return insurance among e-sellers. Some proactively offer this insurance to consumers, while others leave the decision up to them. For example, the Vancl flagship store on the T-mall platform offers return insurance, whereas the Hailan Home flagship store does not (as shown in Fig 1). Additionally, many e-platforms selectively extend return insurance to members who make purchases on the platform, excluding cases where e-sellers independently offer return insurance. For instance, JD.com offers limited free returns to JD Plus members, and T-mall offers free return protection cards to consumers who have physically transacted on T-mall, filled in personal information, and activated their membership. These cardholders are entitled to a certain number of free return shipments per month. The divergent return policies implemented by e-platforms and e-sellers pose intriguing research questions: What are the influences from the return insurance of the competition e-seller? Can e-sellers alter consumer purchasing patterns through pricing strategies? Furthermore, what are the implications of the interplay between return insurance offered by the e-platform and e-sellers?

Unfortunately, there are still some gaps in the current research on return insurance. Most of the research on return insurance focuses on a single-channel environment [1–5]. Research on return insurance in a competitive environment mainly explores the strategic choices of competitive e-sellers, rarely considering the impact of commission rates on the strategic choices of e-sellers' return insurance [6,7]. No research has taken the e-platform as the game subject to explore the optimal return policy of the e-platform. The research on the online return policy of the e-platform and its impact on e-sellers is still an unresolved issue. In fact, some e-platforms improve the return service by presenting consumers with equity version of return insurance, shipping insurance cards, etc. in order to attract them to buy. For example, T-mall gives away return insurance to consumers every month based on their consumption amount and other information on the platform. Therefore, it is necessary to explore the return insurance strategy of the e-platform and its impact on the strategies of e-sellers.

In order to address the aforementioned issues and bridge the existing research gaps, we examine a supply chain comprising an e-platform, two competing e-sellers providing imperfect substitutes, and a heterogeneous group of consumers. By constructing eight duopoly game models, we aim to unravel the optimal return insurance policies tailored to different quality e-sellers and the e-platform, along with the pivotal factors influencing these policies, leading to several noteworthy conclusions. Firstly, we find that when the e-platform does not offer return insurance, e-sellers will judge the size of the premium based on factors such as the premium, the commission rates, the return compensation, and return rate, and then choose an appropriate return strategy. Secondly, when the e-platform does not offer return insurance, e-sellers are able to induce consumers to purchase their own return insurance by reducing prices and providing poor return services. Finally, the provision of return insurance by the e-platform will change the choice of e-sellers' return strategies, which is detrimental to the revenue of high-quality e-sellers, and in turn reduces their own revenue.

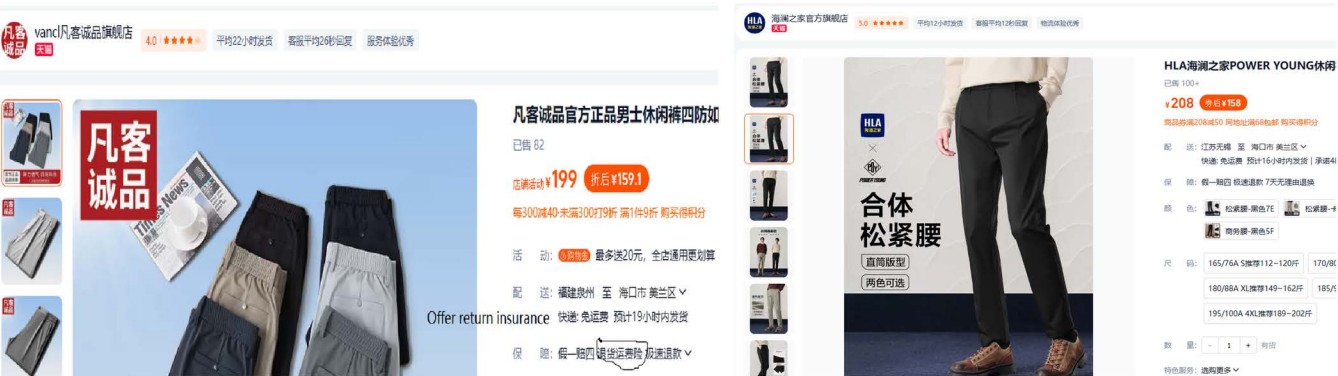

**Fig 1. The Vancl flagship store and the Hailan Home flagship store on T-mall platform.**

Our contributions encompass two crucial aspects. Firstly, in contrast to previous studies that primarily focused on exploring the conditions favoring e-sellers in providing return insurance, we introduce the commission rate as an exogenous variable into the model, thereby enabling a comprehensive analysis of how different return insurance policies impact the optimal pricing strategies, demand, profit, and return insurance policy decisions of two competing e-sellers. Secondly, we relax the earlier assumption that the e-platform operating under an agency sales model is incapable of developing return insurance policies. Instead, we argue that the e-platform within this model function as an independent entity with limited rationality and risk neutrality. Consequently, it is capable of independently devising more favorable return insurance policies. We delve into the implications of such policies on pricing, demand, profit, and return insurance decisions of e-sellers, thereby aligning our analysis more closely with real-world scenarios.

The structure of this study is organized as follows. Section 2 offers a summary of relevant literature. Section 3 details the introduction of the model. Section 4 calculates the optimal pricing strategy, demand, and profit for e-sellers of different qualities without free return insurance on the e-platform. This section also analyzes the return insurance policy choices of high-quality and low-quality e-sellers in a competitive environment. Section 5 extends this analysis to scenarios where the e-platform offers return insurance, examining optimal pricing strategies, demand, profit, and return insurance policies for e-sellers with varying qualities. The section also explores the influence of return insurance offered by the e-platform on the profit of e-sellers with different qualities and the optimal return insurance policy for the e-platform. Section 6 presents the conclusions.

## Literature review

Our research primarily intersects with three research areas: return insurance, consumer return behavior, and return policies.

### Return insurance

Return insurance, designed for goods transportation, involves reimbursement by insurers to retailers or consumers upon product return. On the one hand, some scholars use signaling game theory to explore the signaling function of providing free return insurance [8,9]. For example, Zhang et al. employ signal game theory to demonstrate return insurance as a potent signal for displaying product quality information [8]. Geng and Li explore how heterogeneous consumers interpret return insurance, emphasizing the role of high-quality merchants in adopting free return insurance [9]. On the other hand, some scholars explore whether to provide return insurance and who should provide it [1–7,10–12]. For example, Lin et al. explore whether dual-channel retailers should offer return freight insurance and find that dual-channel retailers should

offer return freight insurance when the production cost, salvage value, and return freight insurance cost of the goods are all high [10]. Zhao and Hu find that it is more effective for retailers to give away return freight insurance from profit and social welfare perspectives [11].

### Consumer return

Pasternack pioneers incorporating return policies into supply chain coordination [13]. Nowadays, the literature related to consumer returns focuses more on the influencing factors of consumer purchase or return decisions, such as the length of the waiting time [14], refund policies [15,16], consumers' perception of the return process [17], and online reviews [18,19]. For example, Xu et al. believed that under the return policy, consumers' valuation of goods depends on the refund amount and the length of time they have to wait after returning the goods, and systematically investigated the impact of the return deadline on consumer behavior [14]. In addition, some scholars have also studied the management decisions related to consumer returns [20–23]. For example, Yan et al. regarded consumer returns as private information and studied the value of product return information to supply chain companies [20]. Wang et al. studied the pricing, ordering, and return policies of option contracts for retailers with customer returns under stochastic demand [21].

### Return policies

Research in this area extends to return channel strategies [24–27], optimal pricing and inventory decisions in omnichannel retail [28–30], and the impact of consumer heterogeneity on retailer services [31]. For example, Zhang et al. study which return policy is more suitable for manufacturers considering consumer utility in a dual-channel supply chain [32]. Lin et al. study the optimal pricing, return policy and return risk value of retailers selling products to customers with uncertain value of products under the two conditions of allowing and not allowing returns [33]. Wang and He discuss whether mass customization retailers should allow returns in dual channels [34]. Ma et al. establish a two-phase model of new product sales by retailers to study the return policy of retailers and its impact [35]. Chen et al. study the return policies and leadership strategies of duopoly retailers with quality differences and find that the application conditions of the full refund policy are still on trial [36]. Qiu et al. construct a robust omnichannel pricing and ordering optimization model, which is applied to two return policies, full refund and non-refund, respectively [37].

Previous studies have laid a solid foundation for subsequent research, and our research differs from previous related literature mainly in two aspects. Firstly, we introduce the commission rate as an exogenous variable to explore its impact on e-sellers' return insurance policies and optimal profits. Secondly, unlike previous research assumptions that e-platforms cannot formulate return strategies under the agency sales model, we assume that the e-platform is an independent entity, and further explore the impact of the e-platform on e-sellers' pricing, demand, profits, and return insurance policies. These additions offer new perspectives to the existing knowledge system.

## Problem description

### Assumption and notation

This paper examines a supply chain consisting of an e-platform, two competitive e-sellers providing imperfect substitutes, with diverse consumer heterogeneity. E-sellers conduct transactions of goods via the e-platform, remitting a predetermined commission based on sales volume [38,39]. Return insurance can be offered by either e-sellers or the e-platform, enabling consumers to choose whether to purchase it or not [39].

The e-platform faces a binary choice: to supply supply return insurance or not. E-sellers are characterized by four cases: NN (neither e-seller offers return insurance), NO (the high-quality e-seller does not offer return insurance, while the low-quality e-seller does), ON (the low-quality e-seller does not offer return insurance, while the high-quality e-seller does), and OO (both e-sellers offer return insurance). The decision sequence entails the e-platform determining return insurance policy, followed by e-sellers determining return policies and retail prices. Consumers make decisions on product

purchase, e-seller choice, and return insurance based on the established policies and prices. Ultimately, consumers decide whether to keep or return the product (Fig 2).

For clarity, Table 1 presents symbols and their meanings, where i represents the return insurance policy of high-quality e-seller and j represents the return insurance policy of low-quality e-seller. N signifies no return insurance, while O denotes the provision of return insurance. For simplification, we refer to the high-quality e-seller as "e-seller 1" and the low-quality e-seller as " e-seller 2".

## Utility of consumers

Assuming neutral consumer risk appetite, normalized to 1 [4], consumers assess the value of goods v, which has heterogeneity and is uniformly distributed from 0 to 1. Product quality is defined as the comprehensive expected utility derived from the consumer's perception of the quality attribute and the matching degree. The matching degree $\alpha_k$ between products sold by e-seller k is considered. We set the time and energy spent by consumers to return goods as $h_0$. Return insurance premiums h are assumed to follow a uniform distribution. The transportation cost for the consumer to return the goods is r.

Therefore, if e-sellers and e-platform don't offer return insurance and consumers don't purchase return insurance, then the consumer's utility is $u_k^{ij} = \alpha_k \left( v - p_k^{ij} \right) - (1 - \alpha_k)(r + h_0)$.

If e-sellers and e-platform don't offer return insurance and consumers buy return insurance, the consumer's utility is $u_k^{ij} = \alpha_k \left( v - p_k^{ij} \right) + (1 - \alpha_k)(h - r + h_0) - i$.

If the e-seller or e-platform offers return insurance, the consumer's utility is $u_k^{ij} = \alpha_k \left( v - p_k^{ij} \right) + (1 - \alpha_k)(h - r + h_0)$.

| E-platform determines return insurance policy and the commission ratio $\beta$ | Two E-sellers determine return insurance policy and retail price $p_k^{ij}$ separately | Consumers decide whether to purchase goods, which retailer to buy from and whether to purchase return insurance | When the consumer has received and experienced the product and decides whether to keep it or return it |
|---|---|---|---|
| stage1 | stage2 | stage3 | stage4 |

**Fig 2. Game sequence diagram.**

**Table 1. Symbols and meanings.**

| Symbols | Meanings |
|---|---|
| $p_k^{ij}$ | The retail price of e-seller k under case ij (decision variable) |
| $r$ | Unit return- freight fee (incurred by consumer) |
| $i$ | Unit return insurance premium purchased by consumer (incurred by consumer) |
| $f_k$ | Unit return insurance premium purchased by e-seller k (incurred by e-seller, $f_1 < f_2$) |
| $v$ | The consumer's valuation of goods |
| $\alpha_k$ | The matching probability of products sold by e-seller k and customer needs ($0 < \alpha_2 < \alpha_1 < 1$) |
| $h$ | Unit refundable freight compensation (obtained by consumer, $0 < \max(f_{Ck}, f_{Ek}) < h \leq r$) |
| $s_k$ | The salvage value of per-unit returned items of e-seller k (incurred by e-seller) |
| $c_k$ | The unit production cost of e-seller k ($0 = c_2 < c_1 < 0.5$) |
| $\beta$ | Proportion of commission (incurred by E-seller) |
| $h_0$ | Unit return hassle cost (incurred by consumer) |

## Profit of e-sellers

E-seller k adopts an agent sales model, selling products to consumers at a price $p_k^{ij}$ through the e-platform. Unit production cost of goods is $c_k$, where $c_1$ and $c_2$ represent the unit production costs of higher and lower quality goods, respectively, with $c_2 < c_1 < 0.5$ [3,39]. Return probability is denoted by unmatching ratio $1 - \alpha_k$. Each returned item incurs losses related to product value and processing costs [4]. Residual value of returned goods is represented by $s_k$. Assume $s_k < c_k < p_k^{ij}$ [13].

Some e-sellers offer return insurance, where the e-seller pays a premium $f_k$ to insure against transportation costs. The insurance compensates consumers with $h$ ($h \leq r$) if goods are returned within a specified time. The return-freight fees and compensation are treated as exogenous variables. Insurance premiums vary based on the return rate of e-sellers, with insurance companies setting higher premiums for e-sellers with higher return rates, then $f_1 < f_2 < r$ [7].

If the e-seller does not offer return insurance, the e-seller's profit is $\pi_k^{ij} = \left[\alpha_k(1-\beta)p_k^{ij} - c_k + (1-\alpha_k)s_k\right]D_k^{ij}$. Otherwise, $\pi_k^{ij} = \left[\alpha_k(1-\beta)p_k^{ij} - c_k + (1-\alpha_k)s_k - f_k\right]D_k^{ij}$.

## Profit of e-platform

The e-platform enables e-sellers to sell goods, charging a commission β based on sales volume [4]. The e-platform may decide in advance whether to offer return insurance to consumers. If return insurance is offered, e-platform's revenue is $\beta\sum_{k=1}^{2}\alpha_k D_k^{ij*}p_k^{ij*} - \sum_{k=1}^{2}D_k^{ij*}f_k$. Otherwise, it is $\beta\sum_{k=1}^{2}\alpha_k D_k^{ij*}p_k^{ij*}$.

## No return insurance from e-platform

### NN case

In this case, neither e-seller offers return insurance. Thus consumers must determine not only from whom to purchase but also whether to purchase return insurance. The indifference point between purchasing high-quality goods and low-quality goods is $v_1^{NN} = \frac{\alpha_1 p_1^{NN} - \alpha_2 p_2^{NN}}{\alpha_1 - \alpha_2} - r - h_0$ when consumers do not purchase return insurance. Conversely, the indifference point between purchasing high-quality goods and low-quality goods shifts to $v_5^{NN} = \frac{\alpha_1 p_1^{NN} - \alpha_2 p_2^{NN}}{\alpha_1 - \alpha_2} - r - h_0 + h$ when consumers purchase return insurance. When there is uncertainty about whether consumers should purchase return insurance, the indifference point between purchasing high-quality goods and purchasing low-quality goods with return insurance is $v_2^{NN} = \frac{\alpha_1 p_1^{NN} - \alpha_2 p_2^{NN} + (1-\alpha_2)h - i}{\alpha_1 - \alpha_2} - r - h_0, 1$, while the indifference point between purchasing high-quality goods with return insurance and purchasing low-quality goods is $v_3^{NN} = \frac{\alpha_1 p_1^{NN} - \alpha_2 p_2^{NN} - (1-\alpha_1)h + i}{\alpha_1 - \alpha_2}$. Furthermore, the consumer will not purchase return insurance for a specific product if and only if $i > (1 - \alpha_k)h$. Utilizing utility theory, we can derive the market segmentation diagram under this case, as depicted in Fig 3.

From Fig 3, we can deduce that the demand function for the high-quality e-seller is $D_1^{NN} = 1 - \frac{\alpha_1 p_1^{NN} - \alpha_2 p_2^{NN}}{\alpha_1 - \alpha_2} + r + h_0 - h^2\frac{2-\alpha_1-\alpha_2}{2}$, while the demand function for the low-quality e-seller is $D_2^{NN} = \frac{\alpha_1\left(p_1^{NN} - p_2^{NN}\right)}{\alpha_1 - \alpha_2} - \frac{r+h_0}{\alpha_2} + \frac{h^2(1-\alpha_1\alpha_2)}{2\alpha_2}$. Additionally, by analyzing the market conditions, we can derive the profit functions for both the high-quality and the low-quality e-sellers as follows:

$$\pi_1^{NN} = D_1^{NN}\left[(1-\beta)\alpha_1 p_1^{NN} - c_1 + (1-\alpha_1)s_1\right] \tag{1}$$

$$\pi_2^{NN} = D_2^{NN}\left[(1-\beta)\alpha_2 p_2^{NN} - c_2 + (1-\alpha_2)s_2\right] \tag{2}$$

To simplify the calculation, let's call $d_i = c_i - s_i(1-\alpha_i)$. This leads to an equilibrium solution for the two e-sellers, which is presented in Lemma 1.

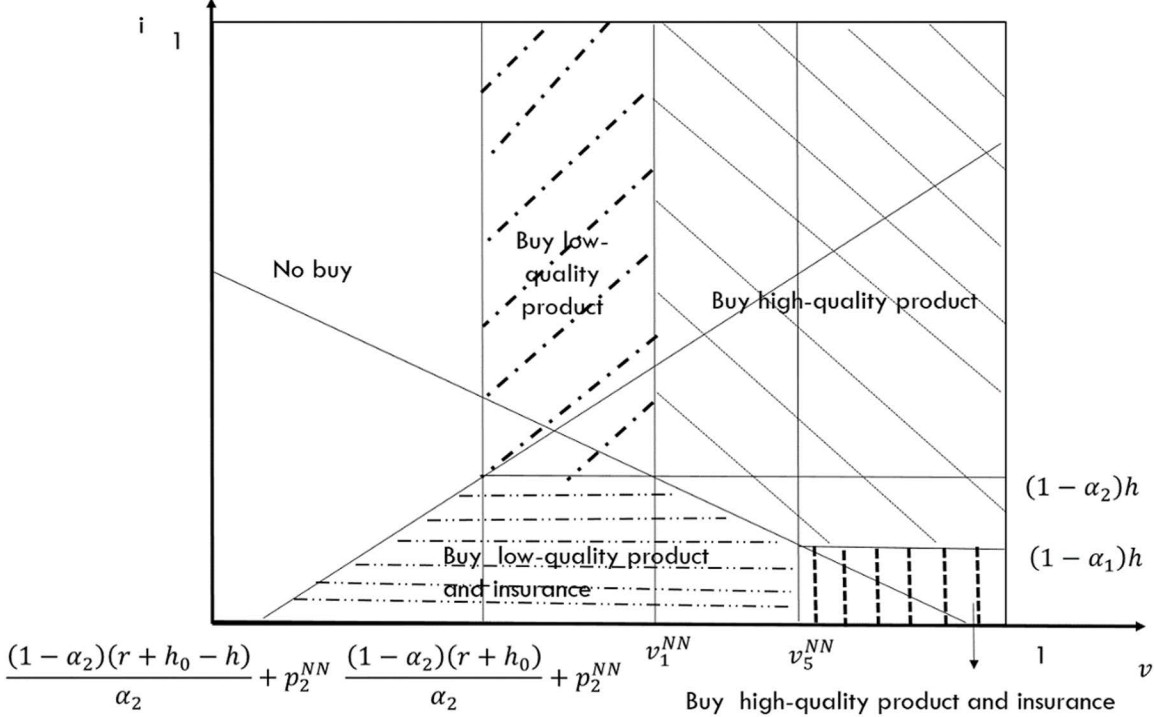

**Fig 3. Demand for incomplete substitutes for NN case that the e-platform doesn't offer return insurance.**

**Table 2. The optimal results of the e-seller in NN case that the e-platform doesn't offer return insurance.**

| E-Seller | Optimal solution |
|---|---|
| High-quality e-seller | $p_1^{NN*} = \dfrac{(\alpha_1-\alpha_2)(1-\beta)\left\{(2\alpha_1-1)\left[2(r+h_0)-h^2(1-\alpha_1)\right]+h^2\alpha_1(\alpha_2-1)+4\alpha_1\right\}+2\alpha_1(2d_1+d_2)}{2(1-\beta)\alpha_1(4\alpha_1-\alpha_2)}$ |
| | $D_1^{NN*} = \dfrac{(1-\beta)(\alpha_1-\alpha_2)\left[h^2(1-\alpha_1)(1-2\alpha_1)-\alpha_1 h^2(1-\alpha_2)-2(r+h_0)(2\alpha_1-1)+4\alpha_1\right]+2\alpha_1(d_2-d_1)-2d_1(\alpha_1-\alpha_2)}{2(1-\beta)(\alpha_1-\alpha_2)(4\alpha_1-\alpha_2)}$ |
| | $\pi_1^{NN*} = \dfrac{\left\{(1-\beta)(\alpha_1-\alpha_2)\left[h^2(1-\alpha_1)(1-2\alpha_1)-\alpha_1 h^2(1-\alpha_2)-2(r+h_0)(2\alpha_1-1)+4\alpha_1\right]+2\alpha_1(d_2-d_1)-2d_1(\alpha_1-\alpha_2)\right\}^2}{4(1-\beta)(4\alpha_1-\alpha_2)^2(\alpha_1-\alpha_2)}$ |
| Low-quality e-seller | $p_2^{NN*} = \dfrac{(1-\beta)(\alpha_1-\alpha_2)\left[2h^2(1-\alpha_2)-h^2\alpha_2(\alpha_1-\alpha_2)+2(r+h_0)(\alpha_2-2)+2\alpha_2\right]+2d_1\alpha_2+4d_2\alpha_1}{2(1-\beta)\alpha_2(4\alpha_1-\alpha_2)}$ |
| | $D_2^{NN*} = \dfrac{\left\{(\alpha_1-\alpha_2)(1-\beta)\left[\alpha_2 h^2(\alpha_2-\alpha_1)+2h^2(1-\alpha_2)+2(r+h_0)(\alpha_2-2)+2\alpha_2\right]+2\alpha_2(d_1+d_2)-4\alpha_1 d_2\right\}\alpha_1}{2(1-\beta)\alpha_2(4\alpha_1-\alpha_2)(\alpha_1-\alpha_2)}$ |
| | $\pi_2^{NN*} = \dfrac{\left\{(\alpha_1-\alpha_2)(1-\beta)\left[\alpha_2 h^2(\alpha_2-\alpha_1)+2h^2(1-\alpha_2)+2(r+h_0)(\alpha_2-2)+2\alpha_2\right]+2\alpha_2(d_1+d_2)-4\alpha_1 d_2\right\}^2\alpha_1}{4(1-\beta)\alpha_2(4\alpha_1-\alpha_2)^2(\alpha_1-\alpha_2)}$ |

**Lemma 1.** Under the NN case, the optimal pricing, demand, and profit of the two e-sellers are as follows, see Table 2 for details.

## NO case

In this case, only the low-quality e-seller refrains from providing return insurance. Consumers must therefore determine whether to purchase high-quality goods and whether to purchase return insurance for those goods. When consumers are

open to purchasing return insurance, the indifference point between purchasing high-quality goods with return insurance and low-quality goods is $v_2^{NO} = \frac{\alpha_1 p_1^{NO} - \alpha_2 p_2^{NO} + i}{\alpha_1 - \alpha_2} - r - h_0 + h$. Conversely, when consumers opt not to buy return insurance, the indifference point between purchasing high-quality goods and low-quality goods is $v_1^{NO} = \frac{\alpha_1 p_1^{NO} - \alpha_2 p_2^{NO} + (1-\alpha_2)h}{\alpha_1 - \alpha_2} - r - h_0$. Furthermore, when a consumer decides to purchase a high-quality product, they will purchase return insurance for that product if $0 < i < (1-\alpha_1)h$. By applying utility theory, we can derive the market segmentation diagram under this case, as depicted in Fig 4.

Based on Fig 4, we can deduce that the demand for the high-quality e-seller is

$D_1^{NO} = 1 - \frac{\alpha_1 p_1^{NO} - \alpha_2 p_2^{NO} + (1-\alpha_2)h}{\alpha_1 - \alpha_2} + r + h_0 + \frac{(1-\alpha_1)^2 h^2}{2(\alpha_1 - \alpha_2)}$, while the demand for the low-quality e-seller is

$D_2^{NO} = \frac{\alpha_1 (p_1^{NO} - p_2^{NO})}{\alpha_1 - \alpha_2} - \frac{r + h_0}{\alpha_2} + \frac{\alpha_1 (1-\alpha_2)h}{\alpha_2(\alpha_1 - \alpha_2)} - \frac{(1-\alpha_1)^2 h^2}{2(\alpha_1 - \alpha_2)}$. Utilizing the information presented in the figure, we can derive the profit functions for both types of e-sellers as follows:

$$\pi_1^{NO} = D_1^{NO} \left[ (1-\beta)\alpha_1 p_1^{NO} + (1-\alpha_1) s_1 - c_1 \right] \tag{3}$$

$$\pi_2^{NO} = D_2^{NO} \left[ (1-\beta)\alpha_2 p_2^{NO} - c_2 + (1-\alpha_2) s_2 - f_2 \right] \tag{4}$$

Lemma 2 can be derived by determining the optimal solution for both e-sellers in the NO scenario, with the objective of maximizing profits.

**Lemma 2.** Under the NO case, the optimal pricing, demand, and profit of the two e-sellers are as follows, see Table 3 for details.

## ON case

In this case, only the high-quality e-seller offers return insurance to consumers. Consumers must decide whether to purchase low-quality goods and whether to purchase return insurance for those goods. When consumers are willing to purchase return insurance, the indifference point between purchasing low-quality goods with return insurance and high-quality goods is $v_2^{ON} = \frac{\alpha_1 p_1^{ON} - \alpha_2 p_2^{ON} - i}{\alpha_1 - \alpha_2} - r - h_0 + h$. Conversely, when consumers choose not to buy return insurance,

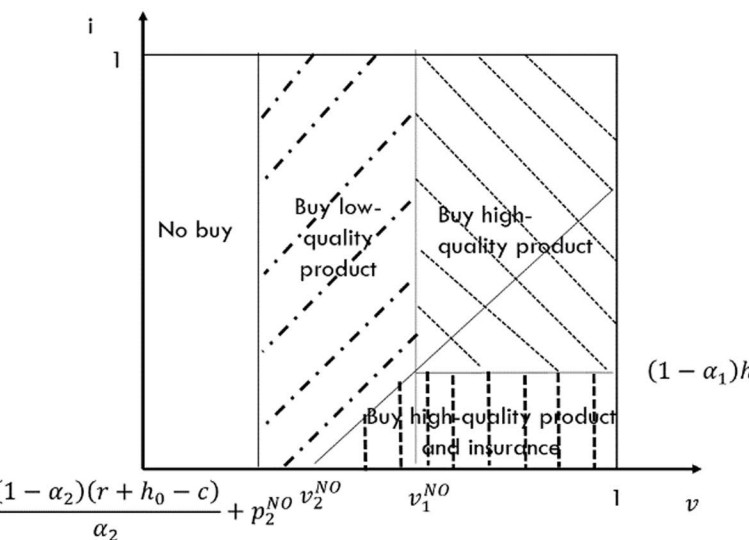

**Fig 4. Demand for incomplete replacement of NO case that the e-platform doesn't offer return insurance.**

**Table 3. Optimal results of the e-seller in NO case that the e-platform doesn't offer return insurance.**

| E-Seller | Optimal solution |
|---|---|
| High-quality e-seller | $p_1^{NO*} = \frac{(1-\beta)(\alpha_1-\alpha_2)\left[h^2(1-\alpha_1)^2+2(r+h_0)(2\alpha_1-1)+4\alpha_1\right]+h^2(1-\beta)(1-\alpha_1)^2\alpha_1+2\alpha_1(2d_1+d_2)-2\alpha_1 h(1-\beta)(1-\alpha_2)+2\alpha_1 f_2}{2(1-\beta)\alpha_1(4\alpha_1-\alpha_2)}$ |
| | $D_1^{NO*} = \frac{(1-\beta)(\alpha_1-\alpha_2)\left[h^2(1-\alpha_1)^2+2(r+h_0)(2\alpha_1-1)+4\alpha_1\right]+h^2(1-\beta)(1-\alpha_1)^2\alpha_1-2(2\alpha_1-\alpha_2)d_1+2\alpha_1 d_2-2\alpha_1 h(1-\beta)(1-\alpha_2)+2\alpha_1 f_2}{2(1-\beta)(4\alpha_1-\alpha_2)(\alpha_1-\alpha_2)}$ |
| | $\pi_1^{NO*} = \frac{\left\{(1-\beta)(\alpha_1-\alpha_2)\left[h^2(1-\alpha_1)^2+2(r+h_0)(2\alpha_1-1)+4\alpha_1\right]+h^2(1-\beta)(1-\alpha_1)^2\alpha_1-2(2\alpha_1-\alpha_2)d_1+2\alpha_1 d_2-2\alpha_1 h(1-\beta)(1-\alpha_2)+2\alpha_1 f\right\}^2}{4(1-\beta)(4\alpha_1-\alpha_2)^2(\alpha_1-\alpha_2)}$ |
| Low-quality e-seller | $p_2^{NO*} = \frac{2h(1-\beta)(2\alpha_1-\alpha_2)(1-\alpha_2)+2(1-\beta)(\alpha_1-\alpha_2)\left[(r+h_0)(\alpha_2-2)+\alpha_2\right]-h^2(1-\beta)(1-\alpha_1)^2\alpha_2+2\alpha_2 d_1+4\alpha_1 f_2+4\alpha_1 d_2}{2(1-\beta)\alpha_2(4\alpha_1-\alpha_2)}$ |
| | $D_2^{NO*} = \frac{\left\{2(2\alpha_1-\alpha_2)[h(1-\beta)(1-\alpha_2)-f_2-d_2]+2(1-\beta)(\alpha_1-\alpha_2)[(r+h_0)(\alpha_2-2)+\alpha_2]-\alpha_2 h^2(1-\beta)(1-\alpha_1)^2+2\alpha_2 d_1\right\}\alpha_1}{2(1-\beta)\alpha_2(4\alpha_1-\alpha_2)(\alpha_1-\alpha_2)}$ |
| | $\pi_2^{NO*} = \frac{\left\{2(2\alpha_1-\alpha_2)[h(1-\beta)(1-\alpha_2)-f_2-d_2]+2(1-\beta)(\alpha_1-\alpha_2)[(r+h_0)(\alpha_2-2)+\alpha_2]-\alpha_2 h^2(1-\beta)(1-\alpha_1)^2+2\alpha_2 d_1\right\}^2\alpha_1}{4(1-\beta)\alpha_2(4\alpha_1-\alpha_2)^2(\alpha_1-\alpha_2)}$ |

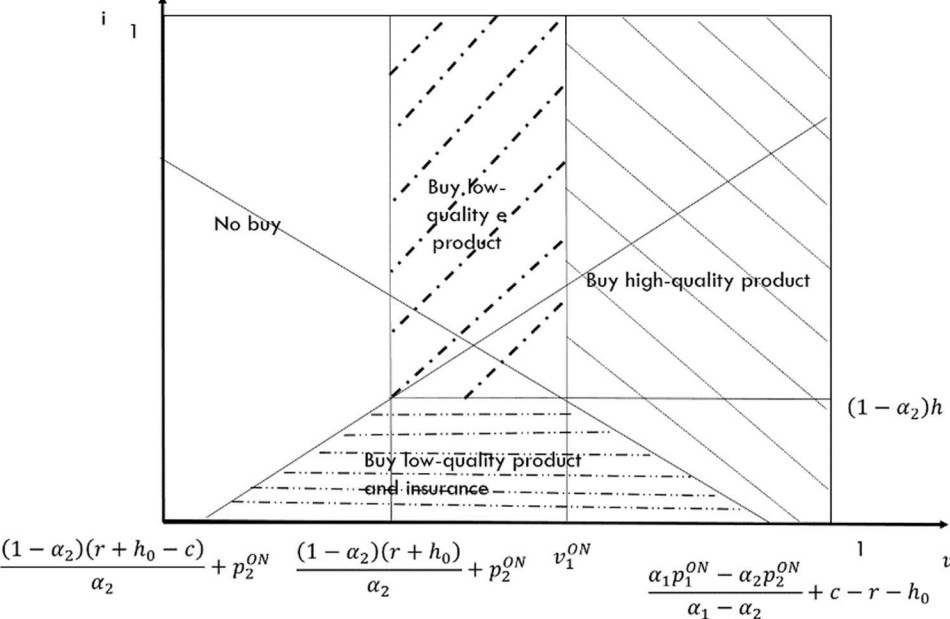

**Fig 5. Demand for incomplete substitutes of ON case that the e-platform doesn't offer return insurance.**

the indifference point between high-quality goods and low-quality goods is $v_1^{ON} = \frac{\alpha_1 p_1^{ON}-\alpha_2 p_2^{ON}-(1-\alpha_1)h(r+h_0)}{\alpha_1-\alpha_2}-r-h_0$. Furthermore, when a consumer opts to purchase a low-quality product, they will purchase return insurance for that product if $0 < i < (1-\alpha_2)h$. By applying utility theory, we can derive the market segmentation diagram in this case, as depicted in Fig 5.

Based on Fig 5, the demand functions for the high-quality e-seller and the low-quality e-seller are

$D_1^{ON} = 1 - \frac{\alpha_1 p_1^{ON}-\alpha_2 p_2^{ON}-(1-\alpha_1)h}{\alpha_1-\alpha_2}+r+h_0-\frac{(1-\alpha_2)^2 h^2}{2(\alpha_1-\alpha_2)}$ and $D_2^{ON} = \frac{\alpha_1\left(p_1^{ON}-p_2^{ON}\right)-(1-\alpha_1)h}{\alpha_1-\alpha_2}-\frac{r+h_0}{\alpha_2}+\frac{\alpha_1(1-\alpha_2)^2 h^2}{2\alpha_2(\alpha_1-\alpha_2)}$ respectively. Utilizing the information presented in the figure, we can derive the profit functions for both types of e-sellers as follows:

$$\pi_1^{ON} = D_1^{ON} \left[ \alpha_1 (1-\beta) p_1^{ON} + (1-\alpha_1) s_1 - c_1 - f_1 \right]$$

(5)

$$\pi_2^{ON} = D_2^{ON} \left[ \alpha_2 (1-\beta) p_2^{ON} - c_2 + (1-\alpha_2) s_2 \right]$$

(6)

Using Formulas (5) and (6), we computed the optimal pricing strategy, demand, and profits for both e-sellers in the ON case. These findings are summarized in Lemma 3.

**Lemma 3.** Under the ON case, the optimal pricing, demand, and profit of the high-quality e-seller and the low-quality e-seller are as follows, see Table 4 for details.

**Table 4. Optimal results of the e-seller in ON case that the e-platform doesn't offer return insurance.**

| E-Seller | Optimal solution |
|---|---|
| High-quality e-seller | $p_1^{ON*} = \dfrac{2(1-\beta)(1-\alpha_1)(2\alpha_1-\alpha_2)h + 2(1-\beta)(\alpha_1-\alpha_2)[2\alpha_1 + (r+h_0)(2\alpha_1-1)] + 2\alpha_1(2d_1+d_2) - h^2(1-\beta)\alpha_1(1-\alpha_2)^2 + 4f_1\alpha_1}{2(1-\beta)\alpha_1(4\alpha_1-\alpha_2)}$ |
| | $D_1^{ON*} = \dfrac{2(1-\beta)(1-\alpha_1)(2\alpha_1-\alpha_2)h + 2(1-\beta)(\alpha_1-\alpha_2)[2\alpha_1 + (r+h_0)(2\alpha_1-1)] - 2(2\alpha_1-\alpha_2)d_1 + 2\alpha_1 d_2 - h^2(1-\beta)\alpha_1(1-\alpha_2)^2 - 4f_1\alpha_1}{2(1-\beta)(4\alpha_1-\alpha_2)(\alpha_1-\alpha_2)}$ |
| | $\pi_1^{ON*} = \dfrac{\left\{ 2(1-\beta)(1-\alpha_1)(2\alpha_1-\alpha_2)h + 2(1-\beta)(\alpha_1-\alpha_2)[2\alpha_1 + (r+h_0)(2\alpha_1-1)] - 2(2\alpha_1-\alpha_2)d_1 + 2\alpha_1 d_2 - h^2(1-\beta)\alpha_1(1-\alpha_2)^2 - 4f_1\alpha_1 \right\}^2}{4(1-\beta)(4\alpha_1-\alpha_2)^2(\alpha_1-\alpha_2)}$ |
| Low-quality e-seller | $p_2^{ON*} = \dfrac{h^2(1-\beta)(1-\alpha_2)^2(2\alpha_1-\alpha_2) + 2(1-\beta)(\alpha_1-\alpha_2)[(r+h_0)(\alpha_2-2)+\alpha_2] - 2\alpha_2(1-\alpha_1)(1-\beta)h + 2\alpha_2(d_1+f_1) + 4\alpha_1 d_2}{2(1-\beta)\alpha_2(4\alpha_1-\alpha_2)}$ |
| | $D_2^{ON*} = \dfrac{\left\{ h^2(1-\beta)(1-\alpha_2)^2(2\alpha_1-\alpha_2) + 2(1-\beta)(\alpha_1-\alpha_2)[(r+h_0)(\alpha_2-2)+\alpha_2] - 2\alpha_2(1-\alpha_1)(1-\beta)h - 2(2\alpha_1-\alpha_2)d_2 + 2\alpha_2(d_1+f_1) \right\}\alpha_1}{2(1-\beta)\alpha_2(4\alpha_1-\alpha_2)(\alpha_1-\alpha_2)}$ |
| | $\pi_2^{ON*} = \dfrac{\left\{ h^2(1-\beta)(1-\alpha_2)^2(2\alpha_1-\alpha_2) + 2(1-\beta)(\alpha_1-\alpha_2)[(r+h_0)(\alpha_2-2)+\alpha_2] - 2\alpha_2(1-\alpha_1)(1-\beta)h - 2(2\alpha_1-\alpha_2)d_2 + 2\alpha_2(d_1+f_1) \right\}^2 \alpha_1}{4(1-\beta)\alpha_2(4\alpha_1-\alpha_2)^2(\alpha_1-\alpha_2)}$ |

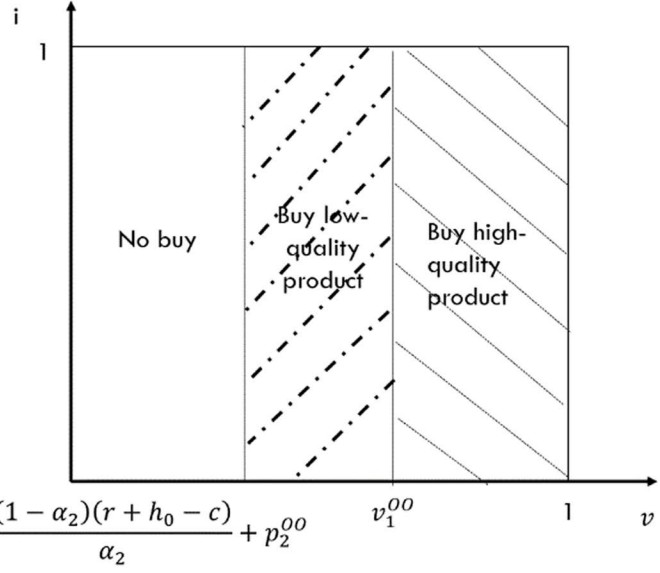

**Fig 6. Demand for incomplete substitutes of OO case that the e-platform doesn't offer return insurance.**

## OO case

In this case, both e-sellers offer return insurance, leaving consumers to solely determine from whom to purchase goods. The threshold at which consumers perceive no difference between purchasing high-quality and low-quality goods is denoted as $v_1^{OO} = \frac{\alpha_1 p_1^{OO} - \alpha_2 p_2^{OO}}{\alpha_1 - \alpha_2} - r - h_0 + h$. By applying utility theory, we can derive the market segmentation diagram for this case, as depicted in Fig 6.

Based on Fig 6, it is evident that the demand functions for the high-quality e-seller and the low-quality e-seller are $D_1^{OO} = 1 - \frac{\alpha_1 p_1^{OO} - \alpha_2 p_2^{OO}}{\alpha_1 - \alpha_2} + r + h_0 - h$ and $D_2^{OO} = \frac{\alpha_1 \left(p_1^{OO} - p_2^{OO}\right)}{\alpha_1 - \alpha_2} + \frac{h - r - h_0}{\alpha_2}$ respectively. Utilizing the information presented in the figure, we can derive the profit functions for both types of e-sellers as follows:

$$\pi_1^{OO} = D_1^{OO}\left[\alpha_1(1-\beta)p_1^{OO} + (1-\alpha_1)s_1 - c_1 - f_1\right] \tag{7}$$

$$\pi_2^{OO} = D_2^{OO}\left[\alpha_2(1-\beta)p_2^{OO} + (1-\alpha_2)s_2 - c_2 - f_2\right] \tag{8}$$

The maximization criterion is used to calculate the optimal pricing, demand and profit of the two e-sellers in the case OO according to Formulas (7)-(8), and Lemma 4 is obtained.

**Lemma 4.** Under the OO case, the optimal pricing, demand, and profit of the high-quality e-seller and the low-quality e-seller are as follows, see Table 5 for details.

## Comparative analysis

In general, the decision of whether e-sellers adopt a return insurance policy primarily depends on whether this policy can bring them additional benefits. This consideration inevitably affects their pricing decisions and product demand.

Firstly, we compare the price, demand, and profit of the high-quality e-seller in the ON and NN cases, as well as those of the low-quality e-seller in the NO and NN cases, to derive Proposition 1.

**Proposition 1.** Providing return insurance inevitably leads to higher prices for e-sellers. The demand and profits of an e-seller will only increase if the premiums are lower, which in turn will lead to lower prices, demand, and profits for the e-seller who does not offer return insurance.

The premium is closely related to the return rate, with a lower premium indicating a higher product fitness. Consumers are more willing to pay higher prices for goods that are more suitable and offer high-quality return services. Therefore, the

**Table 5. Optimal results of e-sellers in OO case that the e-platform doesn't offer return insurance.**

| E-Seller | Optimal solution |
|---|---|
| High-quality e-seller | $p_1^{OO*} = \frac{(1-\beta)(\alpha_1-\alpha_2)[(2\alpha_1-1)(r+h_0-h)+2\alpha_1]+(2d_1+2f_1+d_2+f_2)\alpha_1}{(1-\beta)\alpha_1(4\alpha_1-\alpha_2)}$ |
|  | $D_1^{OO*} = \frac{(1-\beta)(\alpha_1-\alpha_2)[(2\alpha_1-1)(r+h_0-h)+2\alpha_1]+\alpha_1(d_2+f_2)-(2\alpha_1-\alpha_2)(d_1+f_1)}{(1-\beta)(4\alpha_1-\alpha_2)(\alpha_1-\alpha_2)}$ |
|  | $\pi_1^{OO*} = \frac{\{(1-\beta)(\alpha_1-\alpha_2)[(2\alpha_1-1)(r+h_0-h)+2\alpha_1]+\alpha_1(d_2+f_2)-(2\alpha_1-\alpha_2)(d_1+f_1)\}^2}{(1-\beta)(4\alpha_1-\alpha_2)^2(\alpha_1-\alpha_2)}$ |
| Low-quality e-seller | $p_2^{OO*} = \frac{(1-\beta)(\alpha_1-\alpha_2)[(r+h_0-h)(\alpha_2-2)+\alpha_2]+\alpha_2(d_1+f_1)+2\alpha_1(f_2+d_2)}{(1-\beta)\alpha_2(4\alpha_1-\alpha_2)}$ |
|  | $D_2^{OO*} = \frac{\{(1-\beta)(\alpha_1-\alpha_2)[(r-h+h_0)(\alpha_2-2)+\alpha_2]+\alpha_2(d_1+f_1)-(d_2+f_2)(2\alpha_1-\alpha_2)\}\alpha_1}{(1-\beta)\alpha_2(4\alpha_1-\alpha_2)(\alpha_1-\alpha_2)}$ |
|  | $\pi_2^{OO*} = \frac{\{(1-\beta)(\alpha_1-\alpha_2)[(r-h+h_0)(\alpha_2-2)+\alpha_2]+\alpha_2(d_1+f_1)-(d_2+f_2)(2\alpha_1-\alpha_2)\}^2\alpha_1}{(1-\beta)\alpha_2(4\alpha_1-\alpha_2)^2(\alpha_1-\alpha_2)}$ |

e-seller who offers return insurance will benefit from increased demand and profits, while the e-seller who does not offer return insurance must reduce prices to improve their competitiveness.

For a low-quality e-seller, providing return insurance with a higher premium highlights the difference in fitness difference between high-quality and low-quality goods. Despite the insurance, consumers perceive a higher return risk, and the insurance does not alter the probability of a return. Therefore, it is not advisable for informed consumers to purchase low-quality products.

On the other hand, when a high-quality e-seller offers return insurance with a higher premium, it indicates minimal fitness differences between high-quality and low-quality goods. When faced with products of similar fitness, consumers are more inclined to choose the lower-priced, lower-quality option.

Secondly, based on Lemmas 1–4, we can derive the game matrix for the two e-sellers, which is presented in Table 6.

To provide a visual representation, we focus on the variations in the premiums of two e-sellers and the profits of e-sellers of different qualities across distinct intervals under various assumptions. Specifically, Fig 7 depicts the changes in the premium of a high-quality e-seller within the range of $\left(\frac{h(1-\beta)(1-\alpha_1)(2-h+\alpha_1 h)}{2}, f_2\right)$, along with the premium of a low-quality e-seller within the ranges of $\left(f_1, \frac{h(1-\beta)(1-\alpha_2)(2-h+\alpha_2 h)}{2}\right)$ (Fig 7(a)) and $\left(\frac{h(1-\beta)(1-\alpha_2)(2-h+\alpha_2 h)}{2}, r\right)$ (Fig 7(b)), under different cases. On the other hand, Fig 8 illustrates the premium of a high-quality e-seller within the interval of $\left(0, \frac{h(1-\beta)(1-\alpha_1)(2-h+\alpha_1 h)}{2}\right)$, along with the premium of a low-quality e-seller in the ranges of $\left(f_1, \frac{h(1-\beta)(1-\alpha_2)(2-h+\alpha_2 h)}{2}\right)$ (Fig 8(a)) and $\left(\frac{h(1-\beta)(1-\alpha_2)(2-h+\alpha_2 h)}{2}, r\right)$ (Fig 8(b)), under varying assumptions.

As evident from Figs 7 and 8, we find that when the premium of the high-quality e-seller falls within the range of $\left(\frac{h(1-\beta)(1-\alpha_1)(2-h+\alpha_1 h)}{2}, f_2\right)$, the high-quality seller always gains more benefits by not offering return insurance, regardless of whether the low-quality seller provides return insurance. Conversely, when the premium is within the range of $\left(0, \frac{h(1-\beta)(1-\alpha_1)(2-h+\alpha_1 h)}{2}\right)$, offering return insurance is the dominant strategy for the high-quality seller, regardless of whether

**Table 6. A game matrix for high-quality and low-quality e-sellers in scenario that e-platform does not offer return insurance.**

| | | Low-quality e-seller | |
|---|---|---|---|
| | | Not offer | Offer |
| High-quality e-seller | Not offer | $\left(\pi_1^{NN*}, \pi_2^{NN*}\right)$ | $\left(\pi_1^{NO*}, \pi_2^{NO*}\right)$ |
| | Offer | $\left(\pi_1^{ON*}, \pi_2^{ON*}\right)$ | $\left(\pi_1^{OO*}, \pi_2^{OO*}\right)$ |

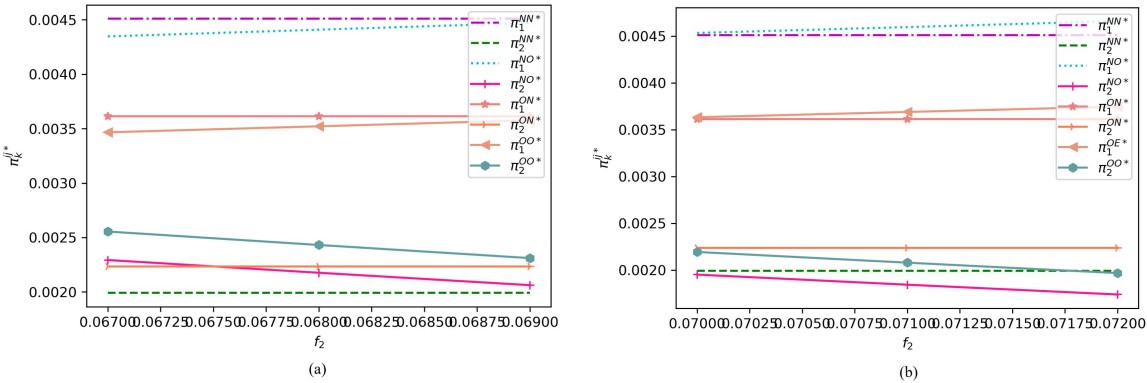

(a) (b)

**Fig 7. Profit changes of the two e-sellers under different cases when** $\frac{h(1-\beta)(1-\alpha_1)(2-h+\alpha_1 h)}{2} < f_1 < f_2$ **(** $\alpha_1 = 0.8$, $\alpha_2 = 0.3$, $c_1 = 0.5$, $c_2 = 0.2$, $\beta = 0.7$, $s_1 = 0.45$, $s_2 = 0.18$, $r = 0.15$, $h = 0.15$, $h_0 = 0.001$, $f_1 = 0.03$**).**

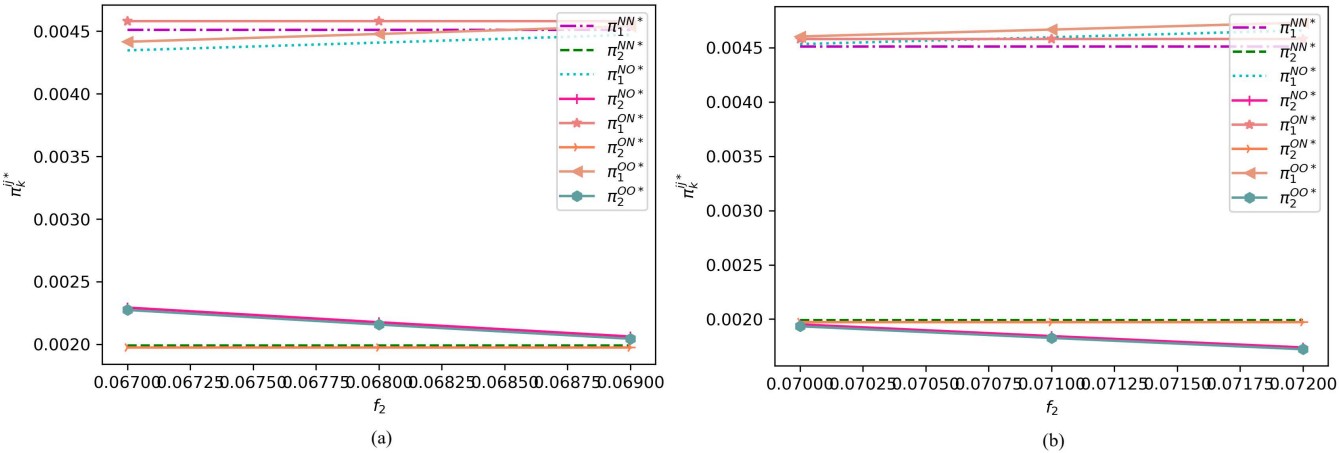

**Fig 8. Profit changes of the two e-sellers under different cases when** $0 < f_1 < \frac{h(1-\beta)(1-\alpha_1)(2-h+\alpha_1 h)}{2}$ ($\alpha_1 = 0.8$, $\alpha_2 = 0.3$, $c_1 = 0.5$, $c_2 = 0.2$, $\beta = 0.7$, $s_1 = 0.45$, $s_2 = 0.18$, $r = 0.15$, $h = 0.15$, $h_0 = 0.001$, $f_1 = 0.02$).

the low-quality seller offers return insurance. Therefore, at higher premiums, the high-quality e-seller may prefers not to offer return insurance to consumers, and vice versa.

To determine the optimal return insurance policy for the high-quality e-seller, we observe that when the premium of the low-quality e-seller is within the $\left(f_1, \frac{h(1-\beta)(1-\alpha_2)(2-h+\alpha_2 h)}{2}\right)$ range, providing return insurance consistently benefits itself, regardless of whether the high-quality seller offers return insurance. Conversely, if the premium is outside this range, not providing return insurance may be more advantageous. Based on these findings, we derive the optimal return policy for both types of e-sellers, as outlined in Proposition 2.

**Proposition 2.** The return policy options of different quality e-sellers are shown in Table 7.

Proposition 2 outlines the return insurance policy choices of two e-sellers under various cases. It is evident that when both e-sellers have excessively high return insurance premiums, neither of them will offer return insurance. In cases where one e-seller has a higher return insurance premium while the other has a lower one, the e-seller with the higher premium will refrain from providing return insurance, whereas the one with the lower premium will offer it. When both e-sellers have relatively low return insurance premiums, both will offer return insurance. Furthermore, combined with Proposition 1, we find that when premiums are relatively high, e-sellers can induce consumers to purchase return insurance on their own at a profit by lowering prices and offering poor return services. According to a search, on the Taobao platform, the official flagship store of Suning Tesco sells a mouse identical to the one sold by Xiaomi's official flagship

**Table 7. Return insurance policies of e-sellers in scenario that e-platform does not offer return insurance.**

| Return insurance policy | Premium for high-quality e-seller $f_1$ | Premium for low-quality e-seller $f_2$ |
|---|---|---|
| NN | $\left(\frac{h(1-\beta)(1-\alpha_1)(2-h+\alpha_1 h)}{2}, f_2\right)$ | $\left(\frac{h(1-\beta)(1-\alpha_2)(2-h+\alpha_2 h)}{2}, r\right)$ |
| NO | $\left(\frac{h(1-\beta)(1-\alpha_1)(2-h+\alpha_1 h)}{2}, f_2\right)$ | $\left(f_1, \frac{h(1-\beta)(1-\alpha_2)(2-h+\alpha_2 h)}{2}\right)$ |
| ON | $\left(0, \frac{h(1-\beta)(1-\alpha_1)(2-h+\alpha_1 h)}{2}\right)$ | $\left(\frac{h(1-\beta)(1-\alpha_2)(2-h+\alpha_2 h)}{2}, r\right)$ |
| OO | $\left(0, \frac{h(1-\beta)(1-\alpha_1)(2-h+\alpha_1 h)}{2}\right)$ | $\left(f_1, \frac{h(1-\beta)(1-\alpha_2)(2-h+\alpha_2 h)}{2}\right)$ |

store, but at a price 21.2 yuan lower than Xiaomi's. However, unlike Xiaomi, Suning Tesco does not offer consumer return insurance. This further demonstrates the accuracy of this conclusion.

This leads to an intriguing conclusion: whether the other party offers return insurance, as long as an e-seller's own return insurance premium is sufficiently low, it will offer return insurance. In essence, whether an e-seller chooses to offer return insurance is solely determined by its own premium and is independent of the other party's return insurance policy. This conclusion aligns with the findings reported by Chen et al. [7]. In reality, some high-quality and low-quality e-sellers adopt the same return policy for similar products. For example, both Estée Lauder and Maybelline have flagship stores on T-mall and offer consumers return insurance. However, some high-quality and low-quality e-sellers adopt different return policies. For instance, on the Taobao platform, the flagship store of The North Face offers consumers return insurance, while the flagship store of Hailan Home does not. These examples suggest that the return insurance policies of e-sellers are not influenced by their competitors.

## The e-platform offers return insurance

In this chapter, we extend the hypothesis to consider a scenario where the e-platform offers return insurance as a complimentary service to consumers, regardless of whether the e-seller offers it or not. Under this setup, the demand function for the high-quality e-seller is denoted as $\widetilde{D_1^{ij}} = 1 - \frac{\alpha_1 \widetilde{p_1^{ij}} - \alpha_2 \widetilde{p_2^{ij}}}{\alpha_1 - \alpha_2} + r + h_0 - h$, while that for the low-quality e-seller is denoted as $\widetilde{D_2^{ij}} = \frac{\alpha_1 \left( \widetilde{p_1^{ij}} - \widetilde{p_2^{ij}} \right)}{\alpha_1 - \alpha_2} + \frac{h - r - h_0}{\alpha_2}$. By analyzing these demand functions, we aim to understand how the availability of free return insurance affects consumer behavior and, subsequently, the profitability of e-sellers. This extended hypothesis allows us to explore new strategies and optimal decisions for e-sellers in a more comprehensive market setup.

## NN case

In Case NN, where neither e-seller offers return insurance but the e-platform offers it to consumers as a complimentary service, the profit expressions for the high-quality and low-quality e-sellers are as follows:

According to the profit maximization criterion, the optimal pricing, demand, and profit of the two e-sellers under this case can be derived, and Lemma 5 can be formulated.

$$\widetilde{\pi_1^{NN}} = \widetilde{D_1^{NN}} \left[ (1-\beta)\alpha_1 \widetilde{p_1^{NN}} - c_1 + (1-\alpha_1)s_1 \right]$$

(9)

**Table 8. Optimal results of e-sellers in NN case that the e-platform may offer return insurance.**

| E-Seller | Optimal solution |
|---|---|
| High-quality e-seller | $\widetilde{p_1^{NN*}} = \frac{(1-\beta)(\alpha_1-\alpha_2)[(1-2\alpha_1)(h-r-h_0)+2\alpha_1]+2\alpha_1 d_1+\alpha_1 d_2}{\alpha_1(1-\beta)(4\alpha_1-\alpha_2)}$ $\widetilde{D_1^{NN*}} = \frac{(-2\alpha_1+\alpha_2)d_1+\alpha_1 d_2+(1-\beta)(\alpha_1-\alpha_2)[(1-2\alpha_1)(h-r-h_0)+2\alpha_1]}{(1-\beta)(4\alpha_1-\alpha_2)(\alpha_1-\alpha_2)}$ $\widetilde{\pi_1^{NN*}} = \frac{\left\{(-2\alpha_1+\alpha_2)d_1+\alpha_1 d_2+(1-\beta)(\alpha_1-\alpha_2)[(1-2\alpha_1)(h-r-h_0)+2\alpha_1]\right\}^2}{(1-\beta)(4\alpha_1-\alpha_2)^2(\alpha_1-\alpha_2)}$ |
| Low-quality e-seller | $\widetilde{p_2^{NN*}} = \frac{(1-\beta)(\alpha_1-\alpha_2)[(2-\alpha_2)(h-r-h_0)+\alpha_2]+(1-\beta)\alpha_2(\alpha_1-\alpha_2)+2\alpha_1 d_2+\alpha_2 d_1}{\alpha_2(1-\beta)(4\alpha_1-\alpha_2)}$ $\widetilde{D_2^{NN*}} = \frac{\left\{(-2\alpha_1+\alpha_2)d_2+\alpha_2 d_1+(1-\beta)(\alpha_1-\alpha_2)[(2-\alpha_2)(h-r-h_0)+\alpha_2]+(1-\beta)\alpha_2(\alpha_1-\alpha_2)\right\}\alpha_1}{(1-\beta)\alpha_2(4\alpha_1-\alpha_2)(\alpha_1-\alpha_2)}$ $\widetilde{\pi_2^{NN*}} = \frac{\left\{(-2\alpha_1+\alpha_2)d_2+\alpha_2 d_1+(1-\beta)(\alpha_1-\alpha_2)[(2-\alpha_2)(h-r-h_0)+\alpha_2]+(1-\beta)\alpha_2(\alpha_1-\alpha_2)\right\}^2\alpha_1}{(1-\beta)\alpha_2(4\alpha_1-\alpha_2)^2(\alpha_1-\alpha_2)}$ |

$$\pi_2^{\widetilde{NN}} = D_2^{\widetilde{NN}} \left[ (1-\beta)\alpha_2 p_2^{\widetilde{NN}} - c_2 + (1-\alpha_2)s_2 \right]$$

(10)

**Lemma 5.** In the case NN, the optimal pricing, demand and profit of high-quality e-seller and low-quality e-seller are as follows, see Table 8 for details.

### NO case

In Case NO, the high-quality e-seller does not offer return insurance, while the low-quality e-seller offers return insurance to consumers. At this time, the profit expressions of the high-quality e-seller and the low-quality e-seller are as follows:

$$\pi_1^{\widetilde{NO}} = D_1^{\widetilde{NO}} \left[ (1-\beta)\alpha_1 p_1^{\widetilde{NO}} - c_1 + (1-\alpha_1)s_1 \right]$$

(11)

$$\pi_2^{\widetilde{NO}} = D_2^{\widetilde{NO}} \left[ (1-\beta)\alpha_2 p_2^{\widetilde{NO}} - c_2 + (1-\alpha_2)s_2 \right]$$

(12)

According to the profit maximization criterion, the optimal pricing, demand, and profit of the two e-sellers under this case can be derived, and Lemma 6 can be formulated.

**Lemma 6.** In the NO case, the optimal demand and profit of high-quality e-seller and low-quality e-seller are as follows, see Table 9 for details.

### ON case

In Case ON, the high-quality e-seller offers return insurance, while the low-quality e-seller does not offer return insurance to consumers. At this time, the profit expressions of the high-quality e-seller and the low-quality e-seller are as follows:

$$\pi_1^{\widetilde{ON}} = D_1^{\widetilde{ON}} \left[ (1-\beta)\alpha_1 p_1^{\widetilde{ON}} - c_1 + (1-\alpha_1)s_1 \right]$$

(13)

$$\pi_2^{\widetilde{ON}} = D_2^{\widetilde{ON}} \left[ (1-\beta)\alpha_2 p_2^{\widetilde{ON}} - c_2 + (1-\alpha_2)s_2 \right]$$

(14)

**Table 9. Optimal results of e-sellers in NO case that the e-platform may offer return insurance.**

| E-Seller | Optimal solution |
|---|---|
| High-quality e-seller | $p_1^{\widetilde{NO}*} = \frac{(1-\beta)(\alpha_1-\alpha_2)[(1-2\alpha_1)(h-r-h_0)+2\alpha_1]+2\alpha_1 d_1+\alpha_1 d_2+\alpha_1 f_2}{\alpha_1(1-\beta)(4\alpha_1-\alpha_2)}$ <br><br> $D_1^{\widetilde{NO}*} = \frac{(-2\alpha_1+\alpha_2)d_1+\alpha_1 d_2+\alpha_1 f_2+(1-\beta)(\alpha_1-\alpha_2)[(1-2\alpha_1)(h-r-h_0)+2\alpha_1]}{(1-\beta)(4\alpha_1-\alpha_2)(\alpha_1-\alpha_2)}$ <br><br> $\pi_1^{\widetilde{NO}*} = \frac{\{(-2\alpha_1+\alpha_2)d_1+\alpha_1 d_2+\alpha_1 f_2+(1-\beta)(\alpha_1-\alpha_2)[(1-2\alpha_1)(h-r-h_0)+2\alpha_1]\}^2}{(1-\beta)(4\alpha_1-\alpha_2)^2(\alpha_1-\alpha_2)}$ |
| Low-quality e-seller | $p_2^{\widetilde{NO}*} = \frac{(1-\beta)(\alpha_1-\alpha_2)[(2-\alpha_2)(h-r-h_0)+\alpha_2]+2\alpha_1 f_2+2\alpha_1 d_2+\alpha_2 d_1}{\alpha_2(1-\beta)(4\alpha_1-\alpha_2)}$ <br><br> $D_2^{\widetilde{NO}*} = \frac{\{(-2\alpha_1+\alpha_2)(d_2+f_2)+\alpha_2 d_1+(1-\beta)(\alpha_1-\alpha_2)[(2-\alpha_2)(h-r-h_0)+\alpha_2]\}\alpha_1}{(1-\beta)\alpha_2(4\alpha_1-\alpha_2)(\alpha_1-\alpha_2)}$ <br><br> $\pi_2^{\widetilde{NO}*} = \frac{\{(-2\alpha_1+\alpha_2)(d_2+f_2)+\alpha_2 d_1+(1-\beta)(\alpha_1-\alpha_2)[(2-\alpha_2)(h-r-h_0)+\alpha_2]\}^2\alpha_1}{(1-\beta)\alpha_2(4\alpha_1-\alpha_2)^2(\alpha_1-\alpha_2)}$ |

**Table 10. Optimal results of e-sellers in ON case that the e-platform may offer return insurance.**

| E-Seller | Optimal solution |
|---|---|
| High-quality e-seller | $\widetilde{p_1^{ON*}} = \dfrac{(1-\beta)(\alpha_1-\alpha_2)[(1-2\alpha_1)(h-r-h_0)+2\alpha_1]+2\alpha_1 d_1+\alpha_1 d_2+\alpha_1 f_1}{\alpha_1(1-\beta)(4\alpha_1-\alpha_2)}$ <br><br> $\widetilde{D_1^{ON*}} = \dfrac{(-2\alpha_1+\alpha_2)(d_1+f_1)+\alpha_1 d_2+(1-\beta)(\alpha_1-\alpha_2)[(1-2\alpha_1)(h-r-h_0)+2\alpha_1]}{(1-\beta)(4\alpha_1-\alpha_2)(\alpha_1-\alpha_2)}$ <br><br> $\widetilde{\pi_1^{ON*}} = \dfrac{\{(-2\alpha_1+\alpha_2)(d_1+f_1)+\alpha_1 d_2+(1-\beta)(\alpha_1-\alpha_2)[(1-2\alpha_1)(h-r-h_0)+2\alpha_1]\}^2}{(1-\beta)(4\alpha_1-\alpha_2)^2(\alpha_1-\alpha_2)}$ |
| Low-quality e-seller | $\widetilde{p_2^{ON*}} = \dfrac{(1-\beta)(\alpha_1-\alpha_2)[(2-\alpha_2)(h-r-h_0)+\alpha_2]+2\alpha_1 d_2+\alpha_2(d_1+f_1)}{\alpha_2(1-\beta)(4\alpha_1-\alpha_2)}$ <br><br> $\widetilde{D_2^{ON*}} = \dfrac{\{(-2\alpha_1+\alpha_2)d_2+\alpha_2(d_1+f_1)+(1-\beta)(\alpha_1-\alpha_2)[(2-\alpha_2)(h-r-h_0)+\alpha_2]\}\alpha_1}{(1-\beta)\alpha_2(4\alpha_1-\alpha_2)(\alpha_1-\alpha_2)}$ <br><br> $\widetilde{\pi_2^{ON*}} = \dfrac{\{(-2\alpha_1+\alpha_2)d_2+\alpha_2(d_1+f_1)+(1-\beta)(\alpha_1-\alpha_2)[(2-\alpha_2)(h-r-h_0)+\alpha_2]\}^2\alpha_1}{(1-\beta)(4\alpha_1-\alpha_2)^2(\alpha_1-\alpha_2)\alpha_2}$ |

According to the profit maximization criterion, the optimal pricing, demand, and profit of the two e-sellers under this case can be derived, and Lemma 7 can be formulated.

**Lemma 7.** In the ON case, the optimal demand and profit of high-quality e-seller and low-quality e-seller are as follows, see Table 10 for details.

## OO case

In the OO case, both e-sellers offer return insurance to consumers. Under this case, the profit functions, optimal pricing strategies, and demand patterns for both the high-quality and low-quality e-sellers remain unchanged from the previous chapter's analysis. Therefore, there is no need to elaborate further on these details here.

## Comparative analysis

Next, we compare and analyze the equilibrium solutions for e-sellers when considering the return insurance offered by the e-platform. Furthermore, we evaluate the optimal profit for the e-platform under different return insurance policies. Through this analysis, we aim to determine the optimal return insurance policy for e-sellers when the e-platform offers return insurance, as well as identify the optimal return insurance policy for the e-platform itself.

First, we compare the optimal price, demand, and profit of the two e-sellers in the ON and NO cases with those in NN case, and then we obtain Proposition3.

**Proposition 3.** When the e-platform offers return insurance, the e-seller's return insurance will reduce its own demand and profits and increase the demand and profits of the other party, resulting in higher prices for both parties.

Proposition 3 illustrates the consequences of an e-seller's decision to offer return insurance on pricing, demand, and profits for both parties involved, considering the case where the e-platform offers return insurance as a complimentary service.

**Table 11. A game matrix for high-quality and low-quality e-sellers in scenario that e-platform offers return insurance.**

| | | Low-quality e-seller | |
|---|---|---|---|
| | | **No offer** | **Offer** |
| High-quality e-seller | No offer | $\left(\widetilde{\pi_1^{NN*}},\widetilde{\pi_2^{NN*}}\right)$ | $\left(\widetilde{\pi_1^{NO*}},\widetilde{\pi_2^{NO*}}\right)$ |
| | Offer | $\left(\widetilde{\pi_1^{ON*}},\widetilde{\pi_2^{ON*}}\right)$ | $\left(\widetilde{\pi_1^{OO*}},\widetilde{\pi_2^{OO*}}\right)$ |

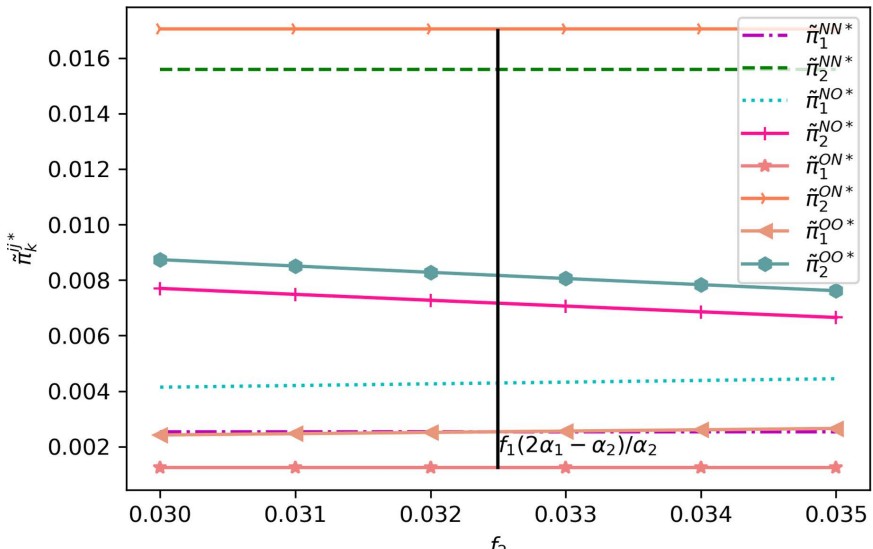

**Fig 9. Profit changes of e-sellers of different quality under the condition that the e-platform offers return insurance** ($\alpha_1 = 0.8, \alpha_2 = 0.3. c_1 = 0.5$, $c_2 = 0.2, \beta = 0.7, s_1 = 0.45, s_2 = 0.18, r = 0.15, h = 0.15, h_0 = 0.001, f_1 = 0.2$).

Firstly, contrary to Proposition 1, the introduction of return insurance by one e-seller leads to an increase in the pricing of the other e-seller. This occurs because the e-platform's provision of return insurance to consumers equalizes the return services offered by both e-sellers, thus providing an opportunity for the non-providing e-seller to raise its prices. Secondly, regardless of the premium level, an e-seller's decision to offer return insurance benefits the other party while potentially damaging its own interests. This occurs as the cost incurred by the providing e-seller in providing return insurance increases, whereas the other e-seller can offer the same return service without bearing the additional cost of purchasing return insurance.

Secondly, according to the profits of the two e-sellers in different cases, we can get the game matrix of the two e-sellers under the scenario that e-platform offers return insurance, as shown in Table 11.

For a more intuitive analysis, we compare the profits in the NO, ON, and OO cases with those in the NN case considering the return insurance offered by the e-platform within the $f_1 \leq f_2 < \frac{(2\alpha_1 - \alpha_2)f_1}{\alpha_1}$ and $\frac{(2\alpha_1 - \alpha_2)f_1}{\alpha_1} \leq f_2 < r$ segments (Fig 9).

As depicted in Fig 9, when the e-platform offers return insurance, it is always beneficial for the high-quality e-seller to refrain from providing return insurance, regardless of its value. Consequently, the high-quality e-seller opts not to offer return insurance. Similarly, at this point, it is always optimal for the low-quality e-seller to forgo offering return insurance, regardless of its value. In fact, since T-mall began providing consumers monthly return insurance as a gift, some e-sellers have stopped giving return insurance to consumers. For example, the flagship store of China's famous men's wear brand Hailan Home and the outlet store of China's famous women's wear brand Wancaoyi no longer offer return insurance to consumers. Based on these observations, we can deduce the optimal return insurance policy for both e-sellers when the e-platform offers return insurance, as stated in Proposition 4.

**Proposition 4.** Neither e-seller will voluntarily offer return insurance for the consumer.

By combining Proposition 2 and 4, we compare and analyze the profits of the e-platform across various return insurance policies. To provide an intuitive understanding, we present profit graphs depicting the e-platform's benefits under different return insurance scenarios in Fig 10. Among them, Fig 10(a), (b), (c) and (d) respectively show the situations where: (a) the e-platform does not offer return insurance, (b) e-sellers do not offer return insurance, (c) only the low-quality e-seller offers it, (d) only the high-quality e-seller offers it, and (e) all offer it. From this, we obtain Observation 1.

**Observation 1.** The e-platform will not voluntarily offer return insurance to consumers.

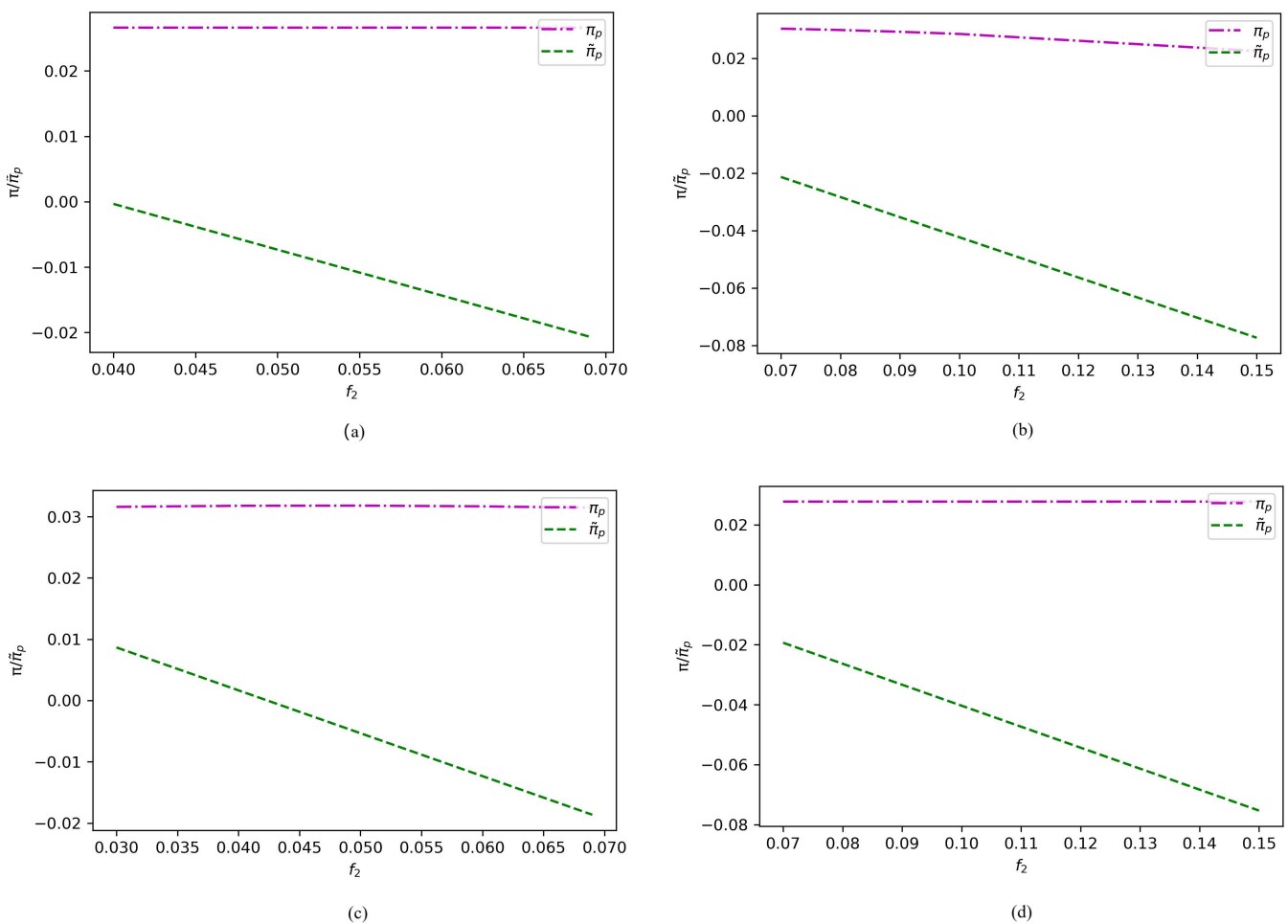

**Fig 10. Profit changes of e-commerce platform before and after providing return insurance** ($\alpha_1 = 0.8, \alpha_2 = 0.3.c_1 = 0.5, c_2 = 0.2, \beta = 0.3, s_1 = 0.45$ ,$s_2 = 0.18, r = 0.15, h = 0.15, h_0 = 0.001$).

Observation 1 indicates that it is difficult for e-platform to gain more profits by voluntarily providing consumers with return insurance. The main reasons are as follows. Firstly, if the e-platform offers return insurance, the two e-sellers will not voluntarily offer return insurance to consumers, which means that the e-platform will have to bear the insurance premiums of the two e-sellers, and the cost of the e-platform will be significantly increased. Secondly, combined with Lemmas 1–8, it can be seen that the demand of high-quality e-seller may increase or decrease with the increase in the return freight fee. Thus, providing return insurance on the e-platform does not necessarily increase the demand for the high-quality e-seller, which also means that e-platform may not only face higher costs, but also may face lower demand for providing return insurance. Finally, because consumers already know the quality information of the goods in advance, return insurance can only reduce the risk and cost of consumers' return, but cannot effectively improve the suitability of the goods. For consumers who buy low-quality goods, it is difficult to accept to spend more money to obtain better return service. This means that the e-platform may not be able to compensate for the higher cost by raising the price. As shown in <u>Fig 10</u> (a)-(d), providing return insurance by the e-platform may lead to negative profits. This conclusion can well explain the phenomenon that e-platforms such as T-mall and JD.com only offer a limited number of return insurance for members within the effective time.

## Conclusion

In the ever-evolving landscape of e-commerce, our study delves into the prevalent issue of product returns. As a counter-measure, major e-platforms have widely adopted return insurance, and employed various strategies to incentivize consumers. Some e-sellers proactively offer return insurance, while others leave the decision up to consumers. Examining a supply chain with two e-sellers, an e-platform, and diverse consumers, our study constructs eight duopoly game models to uncover the optimal return insurance policies and influencing factors.

Our contributions lie in considering the impact of the commission rate and the e-platform's return insurance policy on e-sellers' return insurance decisions. Contrary to previous assumptions, our research challenges the notion that e-platforms operating under the agency sales model cannot formulate return strategies.

We mainly drew the following conclusions.

Firstly, when the e-platform does not offer return insurance, e-sellers with lower premiums will offer it, influenced by factors such as return rate, commission ratio, and return compensation. The size of the premium affects the likelihood of an e-seller providing return insurance.

Secondly, in cases where e-platform does not offer return insurance, if the premiums are high, e-sellers can reduce the price of goods and choose not to offer return insurance, thereby inducing consumers to purchase return insurance on their own initiative to increase revenue.

Thirdly, changes in the e-platform's return insurance policies impact the policies of both e-sellers. E-sellers do not offer return insurance and it is detrimental to the profitability of high-quality e-sellers when the e-platform offers it. It is most advantageous for the e-platform not to voluntarily offer return insurance.

Our conclusions have certain implications for the e-commerce market. Firstly, based on the conditions for e-sellers to offer return insurance, we recommend that e-sellers should comprehensively consider factors such as the product return rate, return compensation, and the commission rate of e-platforms from the perspectives of themselves, insurance companies, and e- platforms to determine the size of the insurance premium before selecting a return strategy. If providing return insurance is not the most suitable return strategy, e-sellers can reduce product prices to attract consumers to purchase by offering discounts and induce consumers to purchase return shipping insurance on their own to reduce return losses. Secondly, based on the optimal return insurance strategy of the e-platform and its impact on e-sellers, we suggest that the e-platform should play its due role by taking certain measures to increase consumer loyalty without interfering with the market, and should not proactively offer return insurance to consumers.

Our findings have practical implications for both e-sellers and the e-platform, emphasizing the need for tailored return insurance policies and the importance of understanding consumer needs. Future research could explore omni-channel supply chains and delve into the strategic choices of the e-platform under the agency sales model regarding return policies. Overall, our study enhances the understanding in the intricate dynamics of return insurance policies in e-commerce.

## Supporting information

**S1 Appendix.  Proof of Proposition.**
(DOCX)

## Acknowledgments

We are very grateful to other students in the research group for their help and support in the publication of the paper.

## Author contributions

**Writing – original draft:** Chen Zhang.

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
