## [Decision Letter · Decision Letter 0]

19 Jul 2024

Dear Dr. Zhang,

Thank you for submitting your manuscript to PLOS ONE. After careful consideration, we feel that it has merit but does not fully meet PLOS ONE’s publication criteria as it currently stands. Therefore, we invite you to submit a revised version of the manuscript that addresses the points raised during the review process.

We look forward to receiving your revised manuscript.

Kind regards,

Ayatulloh Michael Musyaffi

Academic Editor

PLOS ONE

Journal Requirements:

2. Thank you for stating the following financial disclosure: "Postgraduate Innovative Research Projects of Hainan Province(Qhyb2022-20 and Qhyb2021-19)" 

3. We note that your Data Availability Statement is currently as follows: [All relevant data are within the manuscript and its Supporting Information files]

Reviewers' comments:

Reviewer's Responses to Questions

**Comments to the Author**

1. Is the manuscript technically sound, and do the data support the conclusions?

Reviewer #1: Yes

Reviewer #2: Yes

2. Has the statistical analysis been performed appropriately and rigorously?

Reviewer #1: Yes

Reviewer #2: Yes

3. Have the authors made all data underlying the findings in their manuscript fully available?

Reviewer #1: Yes

Reviewer #2: Yes

4. Is the manuscript presented in an intelligible fashion and written in standard English?

Reviewer #1: Yes

Reviewer #2: Yes

Reviewer #1: In my opinion, overall, the abstract accurately reflects the content of the research article. The Introduction section also clearly states the research problem in the form of a question. However, it would be better if gaps from previous studies on the same topic were included, so that the gaps are not only practical but also from previous research.

In the literature review section, several sources have been included, although there is still room to add more sources to increase the number of references.

The methods and results sections have been explained in detail.

In the conclusion section, add other implications that might be beneficial for stakeholders and policymakers.

Reviewer #2: This article establishes a duopoly game model to find the best return insurance policy and its impact on e-sellers and e-platforms. In addition, the following are some suggestions that can be included in the revision.

(1) The language is colloquial and the English level of the paper needs to be improved.

(2) The introduction section can briefly introduce the current research and hypotheses on return insurance policies.

(3) The literature review can be appropriately expanded.

(4) The format of the paper is not standardized.

(5) Some real business cases can be introduced in the analysis and comparison.

(6) Some specific suggestions or opinions can be put forward in the conclusion.

**Do you want your identity to be public for this peer review?** For information about this choice, including consent withdrawal, please see our Privacy Policy

Reviewer #1: No

Reviewer #2: No

---

## [Author Response · Author response to Decision Letter 1]

5 Aug 2024

Dear Ayatulloh Michael Musyaffi,

Thank you very much for giving us an opportunity to revise our manuscript. We appreciate the editor and reviewers very much for their constructive comments and suggestions on our manuscript entitled “Unlocking the Return Insurance Puzzle in E-Commerce: A Strategic Dance Between E-Sellers and E-Platforms”(PONE-D-24-22624). Those comments are all valuable and very helpful for revising and improving our paper, as well as the important guiding significance to our researches. We have studied comments carefully and have made correction which we hope meet with approval.

First, we will reply to the editor's suggestions one by one.

The author’s answer: Dear Editor, thank you very much for your advice! We have revised the format of the paper according to the requirements in the link.

2. Thank you for stating the following financial disclosure: "Postgraduate Innovative Research Projects of Hainan Province(Qhyb2022-20 and Qhyb2021-19)"

The author’s answer: Dear Editor, thank you very much for your carefulness. After discussion, we have removed the funding with the fund number "Qhyb2021-19". The revised financial disclosure is "Postgraduate Innovative Research Projects of Hainan Province (Qhyb2022-20)". The funder, Zhang Chen, has role in the study design, data collection and analysis, decision to publish, and preparation of the manuscript. We have added the financial disclosure and the role of the funder in the research to the cover letter and highlighted it in yellow. Please refer to the re-uploaded cover letter for details.

3. We note that your Data Availability Statement is currently as follows: [All relevant data are within the manuscript and its Supporting Information files]

The author’s answer: Dear Editor, thank you very much for your carefulness. Upon reconfirmation, the values used to construct all the figures are already included in our manuscript.

Next, the main corrections in the paper and the responds to the reviewer’s comments are as following:

Reviewer #1:

In my opinion, overall, the abstract accurately reflects the content of the research article. The Introduction section also clearly states the research problem in the form of a question. However, it would be better if gaps from previous studies on the same topic were included, so that the gaps are not only practical but also from previous research.

The author’s answer: Dear Reviewer, thank you very much for your recognition and valuable suggestions! Based on your advice, we have added in the introduction section the differences between our manuscript and previous studies on the same topic, and we have used the practical example of T-mall offering return insurance to consumers every month to demonstrate the rationality of our manuscript. The specific revisions are as follows: (please refer to the highlighted paragraph at the end of page 3 in the revised manuscript for details)

“Unfortunately, there are still some gaps in the current research on return insurance. Most of the research on return insurance focuses on a single-channel environment [1-5]. Research on return insurance in a competitive environment mainly explores the strategic choices of competitive e-sellers, rarely considering the impact of commission rates on the strategic choices of e-sellers' return insurance. [6, 7] No research has taken e-platforms as the game subject to explore the optimal return policy of e-platforms. The research on the online return policy of e-platforms and its impact on e-sellers is still an unresolved issue. In fact, some e-platforms improve the return service by presenting consumers with equity version of return insurance, shipping insurance cards, etc. in order to attract them to buy. For example, T-mall gives away return insurance to consumers every month based on their consumption amount and other information on the platform. Therefore, it is necessary to explore the return insurance strategy of e-platforms and its impact on the strategies of e-sellers.”

In the literature review section, several sources have been included, although there is still room to add more sources to increase the number of references.

The author’s answer: Dear Reviewer, thank you very much for your suggestions! In accordance with your suggestions, we have added highly relevant literature to our manuscript. The newly added several sources are as follows (please refer to the highlighted section in the "References" part of the revised manuscript for details):

“1. Li XX, Gao J, Bian YW. Return freight insurance strategies for the online retailer and insurance company. International Journal of Production Economics. 2023; 256:108752. https://doi.org/10.1016/j.ijpe.2022.108752

6. Xiong H, Liu J. Return insurance in a competitive market: Benefiting the high-quality or low-quality retailer?. International Journal of Production Economics. 2023; 255:108719. https://doi.org/10.1016/j.ijpe.2022.108719

9. Geng SD, Li WL. Complimentary Return-Freight Insurance Serves as Quality Signal or Noise?. ACIS 2017 Proceedings. 2017; 1-10. https://aisel.aisnet.org/acis2017/47

10. Lin JX, Choi TM, Kuo YH. Will Providing Return-Freight-Insurances Do More Good than Harm to Dual-Channel E-Commerce Retailers?. European Journal of Operational Research. 2023; 307(3):1225-1239. https://doi.org/10.1016/j.ejor.2022.09.025

11. Zhao XM, Hu SH, Meng XX. Who should pay for return freight in the online retailing? Retailers or consumers. Electronic Commerce Research. 2020; 20(2):427-452. https://link.springer.com/article/10.1007/s10660-019-09360-9

12. Zhang TL, Guo XF, Wu T. An analysis of cross-channel return processing with return-freight insurance for live streaming platforms. Computers & Industrial Engineering. 2022; 174:108805. https://doi.org/10.1016/j.cie.2022.108805

18. Das L, Kunja SR. Why do consumers return products? A qualitative exploration of online product return behaviour of young consumers. Journal of Retailing and Consumer Services. 2024; 78:103770. https://doi.org/10.1016/j.jretconser.2024.103770

21. Wang C, Chen J, Chen X. Pricing and order decisions with option contracts in the presence of customer returns. International Journal of Production Economics. 2017; 193:422-436. https://doi.org/10.1016/j.ijpe.2017.08.011

23. Liu JQ, Yuan R, Feng S. Whether to offer return-freight insurance or open a physical showroom: A strategic analysis of an online retailer’s decisions. Journal of Retailing and Consumer Services. 2023; 75:103498. https://doi.org/10.1016/j.jretconser.2023.103498

24. Jin DL, Huang M. Competing e-tailers’ adoption strategies of buy-online-and-return-in-store service. Electronic Commerce Research and Applications. 2021; 47:101047. https://doi.org/10.1016/j.elerap.2021.101047

25. Yang L, Li XY, Xia Y, Aneja YP. Returns operations in omnichannel retailing with buy-online-and-return-to-store. Omega. 2023; 119:102874. https://doi.org/10.1016/j.omega.2023.102874

26. Huang M, Jin DL. Impact of buy-online-and-return-in-store service on omnichannel retailing: A supply chain competitive perspective. Electronic Commerce Research and Applications. 2020; 41:100977. https://doi.org/10.1016/j.elerap.2020.100977

27. Yan S, Archibald TW, Han XH, Bian YW. Whether to adopt “buy online and return to store” strategy in a competitive market?. European Journal of Operational Research. 2022; 301(3):974-986. https://doi.org/10.1016/j.ejor.2021.11.040”

In addition, in order to better highlight the newly added literature, a description of the new literature has been added to the literature review section (please refer to the "Literature Review" section in the revised manuscript, highlighted in yellow):

1) The "Return Insurance" section has been enriched with descriptions of the research conducted by scholars such as Lin and Zhao:

“For example, Lin et al. explore whether dual-channel retailers should offer return freight insurance and find that dual-channel retailers should offer return freight insurance when the production cost, salvage value, and return freight insurance cost of the goods are all high [10]. Zhao and Hu find that it is more effective for retailers to give away return freight insurance from profit and social welfare perspectives [11].”

2) The "Consumer Return" section has been augmented with descriptions of the research conducted by scholars such as Xu, Yan, and Wang.

“For example, Xu et al. believed that under the return policy, consumers' valuation of goods depends on the return amount and the length of time they have to wait after returning the goods, and systematically investigated the impact of the return deadline on consumer behavior[14]. In addition, some scholars have also studied the management decisions related to consumer returns[20-23]. For example, Yan et al. [2] regarded consumer returns as private information and studied the value of product return information to supply chain companies[20]. Wang et al. (2017) [9] studied the pricing, ordering, and return policies of option contracts for retailers with customer returns under stochastic demand[21].”

The methods and results sections have been explained in detail.

The author’s answer: Dear Reviewer, thank you very much for your recognition!

In the conclusion section, add other implications that might be beneficial for stakeholders and policymakers.

The author’s answer: Dear Reviewer, thank you very much for your suggestions! Based on your comments, additional implications that may be beneficial to stakeholders and policymakers have been included in the conclusion section (please refer to page 29 of the revised manuscript, highlighted in yellow).

1) Based on the previous two conclusions regarding e-sellers' pricing strategies and return insurance policies, this revision proposes suggestions focused on formulating e-sellers' pricing and return insurance policies:

“Firstly, based on the conditions for e-sellers to offer return insurance, we recommend that e-sellers should comprehensively consider factors such as the product return rate, return compensation, and the commission rate of e-platforms from the perspectives of themselves, insurance companies, and e- platforms to determine the size of the insurance premium before selecting a return strategy. If providing return insurance is not the most suitable return strategy, e-sellers can reduce product prices to attract consumers to purchase by offering discounts and induce consumers to purchase return shipping insurance on their own to reduce return losses.”

2) Based on the final conclusion regarding e-platforms' return insurance policies, this revision proposes suggestions focused on e-platforms:

“Secondly, based on the optimal return insurance strategy of e-platforms and its impact on e-sellers, we suggest that e-platforms should play their due role by taking certain measures to increase consumer loyalty without interfering with the market, and should not proactively offer return shipping insurance to consumers.”

Reviewer #2: This article establishes a duopoly game model to find the best return insurance policy and its impact on e-sellers and e-platforms. In addition, the following are some suggestions that can be included in the revision.

The author’s answer: Dear Reviewer, thank you very much for your comments! Next, we will respond to each of your suggestions point by point.

(1) The language is colloquial and the English level of the paper needs to be improved.

The author’s answer: Dear Reviewer, thank you very much for your suggestions! Based on your comments, we sought the help of scholars who have published papers in SCI journals and have revised and polished the entire manuscript. Given that the polishing was extensive and involved numerous revised sentences throughout the text, in order to highlight other more critical suggestions, we have not specifically marked the polished sentences. Below, I will illustrate the changes made to the last paragraph of the Introduction before and after the revision.

Before polishing:

“The remainder of this study is organized as follows. Part 2 reviews pertinent literature, while Part 3 provides a detailed description of the model. Part 4 calculates the optimal pricing strategy, demand, and profit for e-sellers of different qualities without free return insurance on the e-platform. This section also analyzes the return insurance policy choices of high-quality and low-quality e-sellers in a competitive environment. Part 5 extends this analysis to scenarios where the e-platform provides return insurance, examining optimal pricing strategies, demand, profit, and return insurance policies for e-sellers with varying qualities. The section also explores the influence of return insurance provided by the e-platform on the profit of e-sellers with different qualities and the optimal return insurance policy for the e-platform. Finally, Part 6 concludes the study.”

After polishing:

“The structure of this study is organized as follows. Section 2 offers a summary of relevant literature. Section 3 details the introduction of the model. Section 4 calculates the optimal pricing strategy, demand, and profit for e-sellers of different qualities without free return insurance on the e-platform. This section also analyzes the return insurance policy choices of high-quality and low-quality e-sellers in a competitive environment. Section 5 extends this analysis to scenarios where the e-platform offers return insurance, examining optimal pricing strategies, demand, profit, and return insurance policies for e-sellers with varying qualities. The section also explores the influence of return insurance offered by the e-platform on the profit of e-sellers with different qualities and the optimal return insurance policy for the e-platform. Section 6 presents the conclusions.”

(2) The introduction section can briefly introduce the current research and hypotheses on return

---

## [Decision Letter · Decision Letter 1]

25 Sep 2024

Dear Dr. Zhang,

Thank you for submitting your manuscript to PLOS ONE. After careful consideration, we feel that it has merit but does not fully meet PLOS ONE’s publication criteria as it currently stands. Therefore, we invite you to submit a revised version of the manuscript that addresses the points raised during the review process.

We recommend that it should be revised taking into account the changes requested by the reviewers. Since the requested changes includes Minor Revision, the revised manuscript will undergo the next round of review by the same reviewers or only by the Academic Editor.

We look forward to receiving your revised manuscript.

Kind regards,

Baogui Xin, Ph.D.

Academic Editor

PLOS ONE

Journal Requirements:

Reviewers' comments:

Reviewer's Responses to Questions

**Comments to the Author**

Reviewer #1: All comments have been addressed

Reviewer #2: (No Response)

2. Is the manuscript technically sound, and do the data support the conclusions?

Reviewer #1: Yes

Reviewer #2: Yes

3. Has the statistical analysis been performed appropriately and rigorously?

Reviewer #1: Yes

Reviewer #2: Yes

4. Have the authors made all data underlying the findings in their manuscript fully available?

Reviewer #1: Yes

Reviewer #2: Yes

5. Is the manuscript presented in an intelligible fashion and written in standard English?

Reviewer #1: Yes

Reviewer #2: Yes

Reviewer #1: After thoroughly reviewing the article, I find that it presents a well-structured and valuable contribution to the field. The research addresses an important and timely issue with a solid theoretical foundation and a well-executed methodology. The discussion is insightful and offers practical implications that can benefit both academics and practitioners.

Reviewer #2: 1、Consistency in Terminology: Ensure consistent use of singular or plural forms. For instance, "E-platforms that do offer" should be "E-platform that offer.".

2、Unify the line spacing and paragraph format of the entire article, for example, all paragraphs have the first line indented.

3、In "assume ( c2 < c1 < 0.5 )", it could be clearer if this assumption is explicitly tied to the context of the model (e.g., quality of goods). Consider rephrasing for more straightforward interpretation, such as "where c1and c2 represent the unit production costs of higher and lower quality goods, respectively, with c2 < c1 < 0.5 ."

4、The size of images in the article should be consistent, and the fonts in the images should be consistent with the fonts in the article.

5、Can describe the picture in detail, especially at the intersection.

**Do you want your identity to be public for this peer review?** For information about this choice, including consent withdrawal, please see our Privacy Policy

Reviewer #1: **Yes: ** Ika Febrilia

Reviewer #2: No

---

## [Author Response · Author response to Decision Letter 2]

31 Jan 2025

Dear Baogui Xin,

Thank you very much for giving us an opportunity to revise our manuscript. We appreciate the editor and reviewers very much for their constructive comments and suggestions on our manuscript entitled “Unlocking the Return Insurance Puzzle in E-Commerce: A Strategic Dance Between E-Sellers and E-Platforms”(PONE-D-24-22624). Those comments are all valuable and very helpful for revising and improving our paper, as well as the important guiding significance to our researches. We have studied comments carefully and have made correction which we hope meet with approval.

First, I would like to explain the situation regarding the delay in submitting the revised manuscript and extend my apologies to the editor.

The revised version has been uploaded to the system. Please kindly check and review it. I sincerely apologize for completing the revisions at this late stage and hope to receive the understanding and acceptance of the editorial office.

The delay occurred because, while working on the manuscript, I received a notification from my school regarding changes to the graduation requirements. As a result, even if this submitted paper were to be accepted, it would not fulfill my graduation requirements, which led me to put it aside for a period of time. However, during my subsequent academic research, I still felt the urge to share this research finding with experts for review and, ideally, have it seen by my peers. Therefore, I have thoroughly revised the paper and responded to each of the expert's comments individually.

I hope the editorial office will grant me this opportunity and take another look at the article. Should there be any issues, please feel free to contact me at any time.

Second, we will reply to the journal requirements.

The author’s answer: Dear Editor, thank you for your suggestion! Upon review, the manuscript does not have the aforementioned issues. Thank you once again for your meticulousness.

Third, we will reply to the reviewer’s comments one by one.

Reviewer #1:

After thoroughly reviewing the article, I find that it presents a well-structured and valuable contribution to the field. The research addresses an important and timely issue with a solid theoretical foundation and a well-executed methodology. The discussion is insightful and offers practical implications that can benefit both academics and practitioners.

The author’s answer: Dear Reviewer, I am extremely grateful for your acknowledgment of the revisions made to my manuscript. Once again, thank you for your professional guidance and meticulous suggestions, which have significantly enhanced the structure and content of my manuscript!

Reviewer #2: 1、Consistency in Terminology: Ensure consistent use of singular or plural forms. For instance, "E-platforms that do offer" should be "E-platform that offer.".

The author’s answer: Dear Reviewer, Thank you for your suggestions and attention to detail! Based on your advice, we have revised the third conclusion in the abstract from "E-platforms that do offer" to "E-platform that offer", highlighted the change in yellow. Additionally, we have carefully read through the entire manuscript and conducted a unified check of all relevant terminology appearing in the text. In response to the inconsistency in terminology you pointed out, we have made individual corrections to ensure consistent use of the same terminology throughout the manuscript.

2、Unify the line spacing and paragraph format of the entire article, for example, all paragraphs have the first line indented.

The author’s answer: Dear Reviewer, thank you very much for your valuable comment regarding the suggestion to unify the line spacing and paragraph format throughout the entire manuscript! After carefully studying recently published manuscripts in Plos One, we noticed a significant formatting characteristic: the first paragraph following each heading does not use first-line indentation. To align our manuscript 's format with the style of Plos One, we have made uniform adjustments to the line spacing and paragraph format according to the format of recently published manuscripts in Plos One. Except for the first paragraph after each section heading, all other paragraphs maintain a consistent line spacing and adopt first-line indentation.

3、In "assume ( c2 < c1 < 0.5 )", it could be clearer if this assumption is explicitly tied to the context of the model (e.g., quality of goods). Consider rephrasing for more straightforward interpretation, such as "where c1and c2 represent the unit production costs of higher and lower quality goods, respectively, with c2 < c1 < 0.5 ."

The author’s answer: Dear Reviewer, thank you very much for your valuable suggestion, which we fully agree with! We have added the description "where c1 and c2 represent the unit production costs of higher and lower quality goods, respectively, with c2 < c1 < 0.5" in the last paragraph on page 8 and highlighted it in yellow. The modifications are as follows:

“Unit production cost of goods is c_k,where c_1and c_2 represent the unit production costs of higher and lower quality goods, respectively, with c_2<c_1<0.5.”

4、The size of images in the article should be consistent, and the fonts in the images should be consistent with the fonts in the article.

The author’s answer: Dear Reviewer, thank you for your valuable suggestion! In response to the issue of inconsistent image sizes and font styles between the images and the text in the article, we have conducted a thorough review of all images and the text contained within them. Subsequently, we have made appropriate adjustments to all images in the manuscript to ensure uniformity in both their sizes and fonts, aligning them seamlessly with the overall style of the manuscript.

5、Can describe the picture in detail, especially at the intersection.

The author’s answer: Dear Reviewer, thank you for your suggestion! We have further elaborated on the relevant descriptions. For instance, in the first paragraph on page 11, to further describe the image, we have provided the following explanation based on the suggestion "Can describe the picture in detail, especially at the intersection” (highlighted it in yellow):

“As evident from Figs 7 and 8, we find that when the premium of the high-quality e-seller falls within the range of (h(1-β)(1-α_1 )(2-h+α_1 h)/2,f_2 ), high-quality seller always gain more benefits by not offering return insurance, regardless of whether low-quality seller provide return insurance. Conversely, when the premium is within the range of (0,h(1-β)(1-α_1 )(2-h+α_1 h)/2), offering return insurance is the dominant strategy for high-quality seller, regardless of whether low-quality seller offer return insurance. Therefore, at higher premiums, high-quality e-seller may prefer not to offer return insurance to consumers, and vice versa.”

Final, today is Chinese New Year. Here I wish you a Happy New Year, all the best, prosperity in the Year of the Snake, and joy for your entire family!

Thank you very much for your attention and time. Look forward to hearing from you.

Yours sincerely,

Chen Zhang

28 Jan, 2025

Hainan University

---

## [Decision Letter · Decision Letter 2]

21 Mar 2025

Unlocking the Return Insurance Puzzle in E-Commerce: A Strategic Dance Between E-Sellers and E-Platforms

PONE-D-24-22624R2

Dear Dr. Zhang,

We’re pleased to inform you that your manuscript has been judged scientifically suitable for publication and will be formally accepted for publication once it meets all outstanding technical requirements.

Kind regards,

Baogui Xin, Ph.D.

Academic Editor

PLOS ONE

Additional Editor Comments (optional):

Reviewers' comments:

Reviewer's Responses to Questions

**Comments to the Author**

Reviewer #3: All comments have been addressed

Reviewer #4: All comments have been addressed

2. Is the manuscript technically sound, and do the data support the conclusions?

Reviewer #3: Yes

Reviewer #4: Yes

3. Has the statistical analysis been performed appropriately and rigorously?

Reviewer #3: Yes

Reviewer #4: Yes

4. Have the authors made all data underlying the findings in their manuscript fully available?

Reviewer #3: Yes

Reviewer #4: Yes

5. Is the manuscript presented in an intelligible fashion and written in standard English?

Reviewer #3: Yes

Reviewer #4: Yes

Reviewer #3: This study explores the optimal return insurance policy in e-commerce and its impact on e-sellers and e-platforms, which is not only a topic of significant research value but also highly innovative. It is gratifying that the authors have successfully addressed and resolved the issues raised by previous reviewers, meeting the publication requirements.

Reviewer #4: This study has good academic value and applicability. The abstract content can be further improved, and it is recommended to include the main results and conclusions of the research in the abstract. In addition, improve the image quality of Figure 1, especially by clearly displaying the keyword 'return insurance'.

**Do you want your identity to be public for this peer review?** For information about this choice, including consent withdrawal, please see our Privacy Policy

Reviewer #3: **Yes: ** ZhengXiang Wu

Reviewer #4: No

---

## [Editor Report · Acceptance letter]

PONE-D-24-22624R2

PLOS ONE

Dear Dr. Zhang,

I'm pleased to inform you that your manuscript has been deemed suitable for publication in PLOS ONE. Congratulations! Your manuscript is now being handed over to our production team.

Kind regards,

on behalf of

Professor Baogui Xin

Academic Editor

PLOS ONE